# Mitochondrial ROS production by neutrophils is required for host antimicrobial function against *Streptococcus pneumoniae* and is controlled by A2B adenosine receptor signaling

**Sydney E. Herring[1], Sovathiro Mao[1], Manmeet Bhalla[1], Essi Y. I. Tchalla[1], Jill M. Kramer[2], Elsa N. Bou Ghanem[1]** *

**1** Department of Microbiology and Immunology, School of Medicine, University at Buffalo, Buffalo, New York, United States of America, **2** Department of Oral Biology, School of Dental Medicine, University at Buffalo, Buffalo, New York, United States of America

* Elsaboug@buffalo.edu

## Abstract

Polymorphonuclear cells (PMNs) control *Streptococcus pneumoniae* (pneumococcus) infection through various antimicrobial activities. We previously found that reactive oxygen species (ROS) were required for optimal antibacterial function, however, the NADPH oxidase is known to be dispensable for the ability of PMNs to kill pneumococci. In this study, we explored the role of ROS produced by the mitochondria in PMN antimicrobial defense against pneumococci. We found that the mitochondria are an important source of overall intracellular ROS produced by murine PMNs in response to infection. We investigated the host and bacterial factors involved and found that mitochondrial ROS (MitROS) are produced independent of bacterial capsule or pneumolysin but presence of live bacteria that are in direct contact with PMNs enhanced the response. We further found that MyD88[-/-] PMNs produced less MitROS in response to pneumococcal infection suggesting that released bacterial products acting as TLR ligands are sufficient for inducing MitROS production in PMNs. To test the role of MitROS in PMN function, we used an opsonophagocytic killing assay and found that MitROS were required for the ability of PMNs to kill pneumococci. We then investigated the role of MitROS in host resistance and found that MitROS are produced by PMNs in response to pneumococcal infection. Importantly, treatment of mice with a MitROS scavenger prior to systemic challenge resulted in reduced survival of infected hosts. In exploring host pathways that control MitROS, we focused on extracellular adenosine, which is known to control PMN anti-pneumococcal activity, and found that signaling through the A2B adenosine receptor inhibits MitROS production by PMNs. A2BR[-/-] mice produced more MitROS and were significantly more resistant to infection. Finally, we verified the clinical relevance of our findings using human PMNs. In summary, we identified a novel pathway that controls MitROS production by PMNs, shaping host resistance against *S. pneumoniae*.

**Data Availability Statement:** All relevant data are within the manuscript and its Supporting Information files.

**Funding:** This work supported by National Institute of Health grants R00AG051784 and R01AG068568-01A1 to ENBG. The content is solely the responsibility of the authors and does not necessarily represent the official views of the National Institutes of Health. This work was also supported by American Heart Association Grant number 827322 to MB. The funders had no role in study design, data collection and analysis, decision to publish, or preparation of the manuscript.

**Competing interests:** The authors have declared that no competing interests exist.

## Author summary

Despite the presence of antibiotics and vaccines, *Streptococcus pneumoniae* infections remain a serious cause of mortality and morbidity globally. Understanding protective host responses is key for designing improved therapies against infection. Neutrophils are innate immune cells that are crucial for control of *S. pneumoniae* infection. In this study we explored the mechanisms by which neutrophils kill *S. pneumoniae*. We found that the mitochondria, whose primary role is energy production, also produce reactive oxygen species (ROS) that are critical for the ability of neutrophils to kill *S. pneumoniae*. We explored the bacterial and host factors involved in mitochondrial ROS (MitROS) production by neutrophils. We found that recognition of bacterial products by neutrophils triggers this response and that the host A2B adenosine receptor regulates it. Importantly, MitROS were required for host resistance against *S. pneumoniae*. This study describes a novel pathway that controls anti-microbial responses and can be a future therapeutic target.

## Introduction

*Streptococcus pneumoniae* (pneumococcus) asymptomatically colonizes the nasopharynx in most individuals but can progress to become a lethal pathogen resulting in over 1 million deaths annually worldwide [1]. The most common site of pneumococcal infection is the lung, but certain bacterial strains can also spread systemically, resulting in bacteremia and disseminated infection [1]. Innate immune cells such as polymorphonuclear leukocytes (PMNs), also known as neutrophils, are crucial for control of bacterial numbers early in infection [2,3]. In mouse models, depletion of PMNs prior to infection results in increased bacterial burden in the lungs and blood and neutropenic individuals are at an increased risk of pneumococcal pneumonia [2–5]. These findings highlight the importance of PMNs in host defense against *S. pneumoniae*.

PMNs have several mechanisms by which they can kill bacteria both extracellularly and within phagosomes [3]. One such mechanism is by production of reactive oxygen species (ROS) such as superoxide anion ($O_2\bullet-$) and hydrogen peroxide ($H_2O_2$). These molecules can react with and oxidize many cellular components including lipids, proteins and nucleic acid and therefore can directly damage and kill bacteria [6]. In PMNs, a major source of ROS is the NADPH oxidase, a multiprotein complex that can assemble both at the phagosomal membrane as well as at the PMN surface resulting in ROS production both extracellularly and within the cell [6]. However, the NADPH oxidase is dispensable for host resistance against *S. pneumoniae*. Patients with Chronic Granulomatous Disease (CGD), which is caused by mutations in the different components of the NADPH oxidase complex, lack a functional NADPH oxidase and display defects in oxidative burst [7]. Although these patients are highly susceptible to recurrent infections by several pathogens, *S. pneumoniae* is not one of them [7,8]. Further, mice that lack different components of the NADPH oxidase do not display defects in their ability to control bacterial numbers following pulmonary challenge with *S. pneumoniae* [9,10]. Additionally, pharmacological inhibition of the NADPH oxidase complex does not impair the ability of PMNs to kill *S. pneumoniae* [11,12]. These findings and observations suggest that either ROS production by PMNs was not required for host defense against *S. pneumoniae* or alternatively that there are other non-NADPH oxidase sources of ROS within PMNs that are sufficient to maintain antimicrobial activity. In support of the latter, we previously

found that ROS are in fact required for the ability of PMNs to kill *S. pneumoniae*, as treatment of PMNs with compounds that detoxify or scavenge ROS completely abrogated the ability of these cells to kill bacteria [12]. These findings suggest that alternate cellular sources of ROS, other than the NADPH oxidase, are important for bacterial killing by PMNs.

The mitochondria are an important source of ROS within cells [13]. Mitochondrial ROS (MitROS) are produced when electrons escaping from the electron transport chain are picked up by oxygen [14]. MitROS production in the context of infection has been mostly characterized in macrophages [15]. Several studies have shown that pathogen recognition by Toll-like receptors results in upregulation of MitROS production and/or triggers signaling cascades that result in trafficking of the mitochondria or mitochondrial-derived vesicles containing ROS to the pathogen engulfed within phagosomes [16–19]. MitROS production enhances the microbicidal activity of macrophages against intracellular bacteria such as *Listeria monocytogenes* [19,20], *Mycobacterium tuberculosis* [21] and *Salmonella* [17] as well as extracellular bacteria that have been engulfed such as *Staphylococcus aureus* [16] and *Escherichia coli* [18]. MitROS production in PMNs is not as well-characterized. PMNs have an extensive mitochondrial network [22] and a few studies have found that MitROS are produced in PMNs in response to stimulation with N-Formylmethionine-leucyl-phenylalanine (fMLP) [22,23], release of intracellular $Ca^{2+}$ that typically occurs downstream of pathogen recognition [24], and in response to ER stress triggered by antigen/antibody complexes or *S. aureus* infection [25,26]. MitROS production in PMNs boosted select anti-microbial effector functions and was required for release of primary and secondary granules in response to fMLP [24] and enhanced formation of neutrophil extracellular traps (NETS) in response to a few stimuli [24,25]. Importantly, MitROS were required for efficient NETosis of PMNs from CGD patients [24], demonstrating that in the absence of NADPH oxidase, the mitochondria are an important alternate source of cellular ROS. The importance of MitROS in the ability of PMNs to kill pathogens remains poorly explored.

Although MitROS production has been shown to be important for overall host defense against several pathogens [27], this response can be damaging to the host particularly in organs where inflammation can be detrimental to function such as the lungs [27–31]. In fact, inhibition of the mitochondrial electron transport chain complexes ameliorated lung injury in response to LPS [29,30] and scavenging MitROS from the lungs of influenza A virus infected mice ameliorated pulmonary inflammation [31]. Pathways that regulate MitROS production by PMNs have not been explored. Extracellular adenosine (EAD) is a key regulator of PMN responses during pneumococcal infection [2,12,32–34]. EAD is produced in the extracellular environment as a breakdown product of ATP released from damaged cells where two extracellular enzymes CD39 and CD73 sequentially de-phosphorylate ATP to EAD [35]. EAD can then act on four G-protein coupled receptors, A1, A2A, A2B and A3 that have varying affinities to EAD and are expressed on many cells including PMNs [34,36]. These receptors are also coupled to different G proteins that can inhibit (Gi) or stimulate (Gs) adenylyl cyclase or activate phospholipase C (Gq), and therefore can have opposite effects on downstream signaling and cell function [36]. The higher affinity receptors A1 and A3 are Gi coupled, while the intermediate and low affinity A2A and A2B receptors are Gs coupled [36]. We previously found that extracellular adenosine production by CD73 and signaling via the Gi-coupled A1 receptor is required for host resistance and for the ability of PMNs to kill *S. pneumoniae* [34,37]. However, the role of the other adenosine receptors in pneumococcal infection has not been explored. It was reported that signaling via A2A and A2B inhibit ROS production by PMNs in response to fMLP and cytokine stimulation [38–40]. Whether adenosine receptor signaling regulates MitROS production by PMNs is not known.

In this study we asked the question of whether the mitochondria act as a source of ROS in PMNs infected with *S. pneumoniae* and probed the host pathways controlling this response as well as the role of MitROS in PMN antibacterial function and host resistance to infection. We found that MitROS were produced by PMNs in response to released bacterial products in a MyD88-dependent manner and that signaling via the A2B receptor inhibited this response. Importantly, MitROS were required for the ability of PMNs to kill *S. pneumoniae ex vivo* and for the ability of the host to clear systemic infection. These findings reveal the role of MitROS in shaping host resistance against *S. pneumoniae* and describe a novel pathway that controls this important anti-microbial response.

## Results

### *S. pneumoniae* infection induces mitochondrial ROS production in PMNs

We previously found that ROS were required for PMN antimicrobial activity against *S. pneumoniae*, however the NADPH oxidase was dispensable for that [11,12]. ROS can be produced from non-NADPH oxidase sources within cells including the mitochondria [13]. To test whether MitROS are produced by PMNs in response to infection, we directly measured mitochondrial ROS in a flow cytometry-based assay using MitoSOX, a dye that is specifically targeted to the mitochondria and fluoresces when it is oxidized by superoxide (Fig 1A) [23]. We found that infection of PMNs with serotype 4 *S. pneumoniae* TIGR4 strain triggered MitROS production where we observed a dose-dependent increase in the amount of MitROS produced (geometric MFI, Fig 1B) and a 4-5-fold increase in the percentages of PMNs producing MitROS in response to infection (Fig 1C). To confirm that this method in fact detects ROS produced by the mitochondria, we added MitoTEMPO during infection, a mitochondria-specific superoxide scavenger [41,42]. We found that upon incubation with MitoTEMPO, the amount and percentages of MitROS detected returned to baseline (Fig 1A and 1B), confirming the specificity of the MitoSOX method. We then wanted to determine what was the relative contribution of MitROS to the intracellular ROS pool. To do so, we used a chemiluminescent-based ROS detection assay with luminol, which is cell and mitochondria permeable [43] and detects total intracellular superoxide production [44]. As previously reported [12], we found that *S. pneumoniae* infection rapidly up regulated intracellular ROS production within PMNs (Fig 1D). Addition of MitoTEMPO significantly reduced the total amount of intracellular ROS detected in infected PMNs resulting in a 6-fold reduction at the peak of the response (Fig 1D). Taken together, these findings suggest that the mitochondria are an important source of overall intracellular ROS produced by PMNs in response to *S. pneumoniae* infection.

### Mitochondrial ROS production by PMNs is triggered by released bacterial products and enhanced by contact

To understand what triggers MitROS production, we started by investigating the bacterial factors involved. We first asked if MitROS production was dependent on interaction with live bacteria. Using the MitoSOX flow cytometry-based assay, we measured mitochondrial ROS production in PMNs in response to live, heat killed, or formalin fixed *S. pneumoniae*. We found that killed (by heat and formalin) bacteria failed to induce a robust increase in mitochondrial ROS production (Fig 2A) suggesting that live bacteria and/ or bacterial factors that are heat-labile or sensitive to formalin fixation are required for this response. We then asked whether MitROS production required PMNs to be actively bound by bacteria. To do so, we used GFP-expressing *S. pneumoniae* TIGR4 to infect cells. We then gated on PMNs that bound bacteria (GFP+) or those that had no bacteria actively bound (GFP Neg) (Fig 2B) and

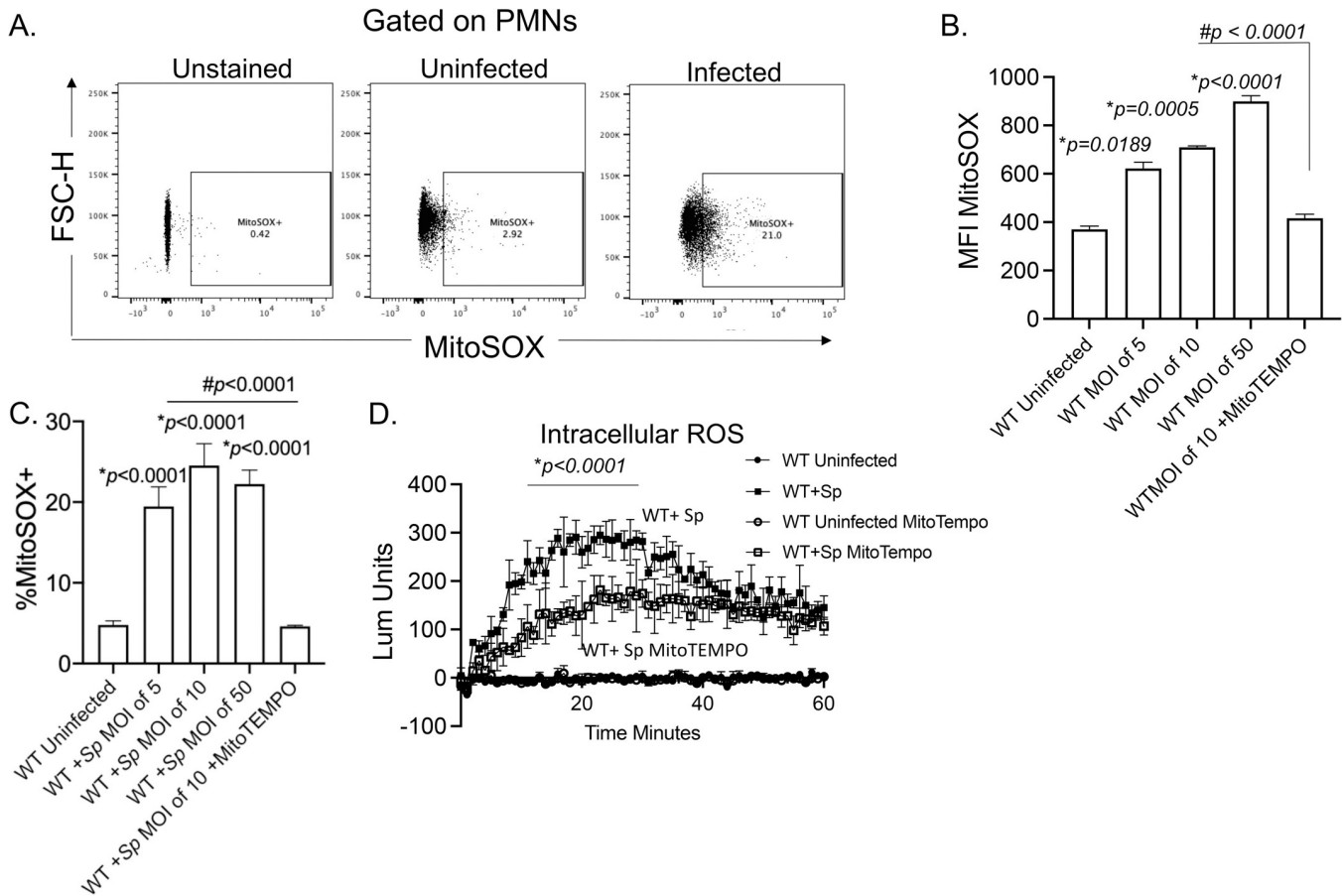

**Fig 1. Mitochondrial ROS are produced by PMNs in response to *Streptococcus pneumoniae* infection.** WT (C57BL/6) bone marrow-derived PMNs were infected with *S. pneumoniae* TIGR4 at various MOIs for 10 minutes and mitochondrial ROS measured using MitoSOX. (A) Gating strategy. (B) The geometric Mean Fluorescent Intensity (MFI) or (C) % of mitochondrial ROS producing cells were determined by flow cytometry. (D) PMNs were treated with vehicle control or the mitochondrial ROS scavenger MitoTEMPO and then mock treated (uninfected) or infected with *S. pneumoniae* (+Sp) TIGR4 at a MOI of 50. Cells were monitored for intracellular ROS production over the course of 60 minutes using chemiluminescent luminol. (B-D) Representative data shown are from 1 out of 5 separate experiments in which n = 3 technical replicates were used per condition. Line and Bar graphs represent the mean +/-SD. (B and C) * indicates significant differences from uninfected controls and # indicates significant differences between the indicated groups as measured by one-way ANOVA followed by Tukey's multiple comparison test. (D) * indicates significant differences between infected groups +/- MitoTEMPO as determined by 2-way ANOVA followed by Tukey's multiple comparisons test.

compared their ability to produce MitROS. We detected similar MitROS production in both PMNs directly interacting with bacteria, and those that had no bacteria bound (Fig 2B). As in the above assay some GFP Neg PMNs may have been in direct contact with bacteria at one point, and to further investigate the requirement of contact, we separated PMNs from the bacteria using a 0.4μm Transwell (Fig 2C). We found that despite having a significant increase in MitROS in PMNs separated from the bacteria by a Transwell compared to uninfected controls, the percentages of directly infected PMNs producing MitROS was 2-fold higher (Fig 2C). As some bacterial products may adhere to the Transwell, we directly tested if factors released by the bacteria were sufficient to induce MitROS production by culturing bacteria in assay buffer and then treating the PMNs with cell-free supernatants. We found that bacterial supernatants were sufficient to trigger MitROS by PMNs (Fig 2D). These findings suggest that secreted/ released bacterial products are sufficient to induce MitROS production by PMNs.

The data above suggest that mitochondrial ROS are triggered in part by released bacterial factor(s) that are deactivated by heat and/ or fixation. *S. pneumoniae* produce pneumolysin

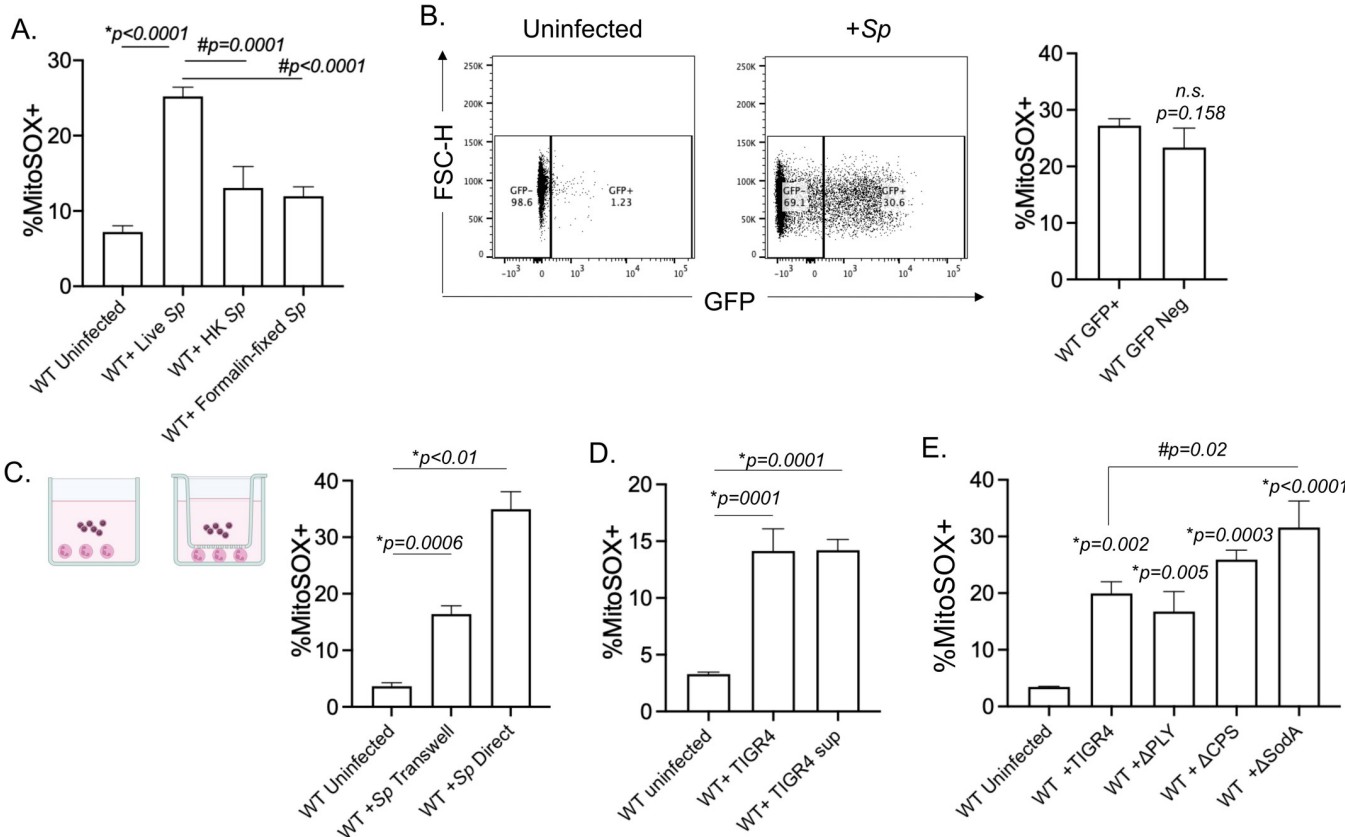

**Fig 2. Bacterial factors required for mitochondrial ROS production by PMNs.** (A) WT (C57BL/6) bone-marrow derived PMNs were mock-treated (Uninfected) or infected with Live (Live Sp), heat-killed (HK Sp) or formalin fixed (Formalin-fixed Sp) *S. pneumoniae* TIGR4 at a MOI of 10 for 10 minutes and mitochondrial ROS was measured using MitoSOX. Data are representative of 1 out of 3 separate experiments in which n = 3 technical replicates were used per condition. (B) WT PMNs were infected with GFP-expressing *S. pneumoniae* TIGR4 at a MOI of 10 for 10 minutes. GFP positive vs. negative populations were gated on and compared for production of mitochondrial ROS using MitoSOX by flow cytometry. Data are representative of 1 of 5 experiments in which n = 3 technical replicates were used per condition. (C) WT PMNs were seeded in 24-well plates and either directly infected with *S. pneumoniae* TIGR4 at a MOI of 10, or were separated from the bacteria by a trans-well. The % of MitoSOX+ cells were determined using flow cytometry. Data are representative from 1 of 3 separate experiments in which n = 3 technical replicates were used per condition. (D) PMNs were either directly infected with *S. pneumoniae* TIGR4 at a MOI of 10, or treated with bacterial supernatant. The % of MitoSOX+ cells were determined using flow cytometry. Data are representative from 1 of 4 separate experiments in which n = 3 technical replicates were used per condition. (E) WT PMNs were mock-treated (Uninfected) or infected at a MOI of 10 for 10 minutes with either wild type *S. pneumoniae* TIGR4 (+TIGR4), a pneumolysin deletion mutant (+ΔPLY), a capsular deletion mutant (+ΔCPS) or bacteria lacking superoxide dismutase (+ΔsodA). The % of MitoSOX+ cells were determined by flow cytometry. Data are representative of 1 out of 5 separate experiments in which n = 3 technical replicates were used per condition. Bar graphs represent the mean +/-SD. * indicates significant differences from uninfected controls and # indicates significant differences between indicated groups as measured by one-way ANOVA followed by Tukey's multiple comparison test. n.s. indicates not significant.

(PLY), a pore forming toxin which is fixation sensitive and heat-labile, is released extracellularly by bacteria [45] and was found to induce respiratory burst in PMNs [46]. Therefore, we tested the ability of a PLY deletion mutant to trigger MitROS. We found that *S. pneumoniae* Δ*PLY* elicited MitROS comparable to wildtype bacteria, indicating that MitROS production by PMNs is independent of PLY (Fig 2E). As interaction with heat-killed bacteria elicited a modest increase in MitROS producing PMNs, we also investigated whether expression of heat-resistant capsular polysaccharides which can be shed by the bacteria [47] are required for MitROS production. To do so, we infected PMNs with a capsule deletion mutant and found that *S. pneumoniae* Δ*cps* elicited MitROS comparable to wildtype bacteria, indicating that MitROS production by PMNs is independent of capsule expression (Fig 2E). Finally, we tested MitROS levels upon infection with *S. pneumoniae* lacking the manganese-dependent

superoxide dismutase ($\Delta sodA$), which is the main enzyme used by these bacteria to degrade superoxide radicals [48]. We found that MitROS producing PMNs were significantly higher upon infection with this bacterial mutant compared to wild type bacteria (Fig 2E), suggesting that *S. pneumoniae* utilizes *sodA* to detoxify MitROS. Overall, these findings suggest that MitROS may be triggered by multiple mechanisms where released bacterial factors are sufficient to elicit MitROS production but presence of live bacteria that are in direct contact with PMNs enhances the response.

## MyD88 signaling contributes to mitochondrial ROS production by PMNs

As MitROS production by PMNs depended on bacterial factors and MitROS production in macrophages is known to depend on TLRs, we next investigated the role of MyD88 in this response. To do so, we measured MitROS production in PMNs isolated from wild type (WT) C57BL/10, MyD88$^{+/-}$, or MyD88$^{-/-}$ mice. We found that pneumococcal infection elicited an increase in the percentage of PMNs producing MitROS compared to uninfected controls in PMNs from all three mouse strains (Fig 3A and 3B). However, there was a significant 3-fold lower response observed in in MyD88$^{-/-}$ PMNs (Fig 3) compare to WT PMNs. These findings indicate that MyD88 is needed for optimal mitochondrial ROS production by PMNs in response to *S. pneumoniae* infection.

## Mitochondrial ROS are required for the ability of PMNs to kill *S. pneumoniae*

We previously found that scavenging/ detoxifying ROS completely abrogates the ability of PMNs to kill *S. pneumoniae* [12]. To determine if mitochondrial ROS were important for the

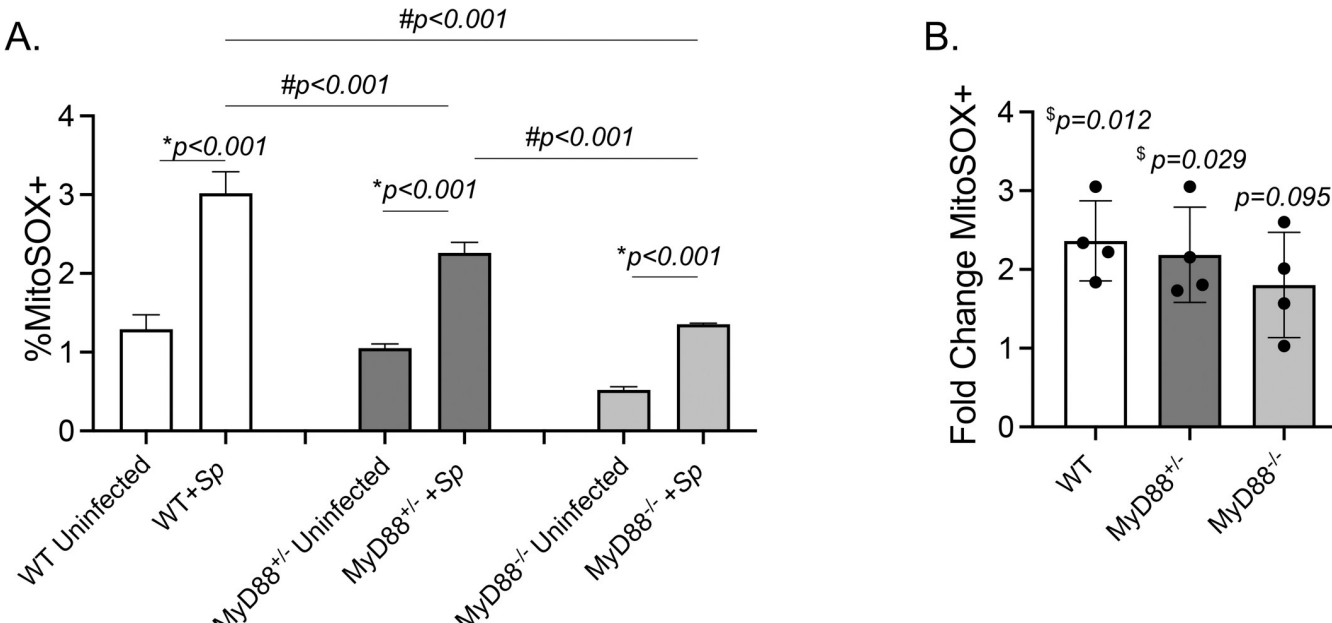

**Fig 3. MyD88 signaling required for mitochondrial ROS production by infected PMNs.** Bone-marrow-derived PMNs were isolated from C57BL10 (WT) (open bars), MyD88$^{+/-}$ (dark bars), and MyD88$^{-/-}$ (grey bars) mice. PMNs were infected with *S. pneumoniae* TIGR4 (+Sp) at a MOI of 10 or mock-treated (Uninfected) for 10 minutes. (A) The % of MitoSOX+ cells were determined using flow cytometry. Data are representative from 1 of 4 separate experiments in which n = 3 technical replicates were used per condition. Bar graphs represent the mean +/-SD. * indicates significant differences from uninfected controls and # indicates significant differences between indicated groups as measured by one-way ANOVA followed by Tukey's multiple comparison test. (B) Fold increases in MitoSOX+ cells upon bacterial infection were calculated by dividing the values of infected conditions by uninfected controls for each mouse strain. Data pooled from four separate experiments (n = 4 mice/ group) are shown. $ indicates significantly different from 1 as measured by one-sample t-test.

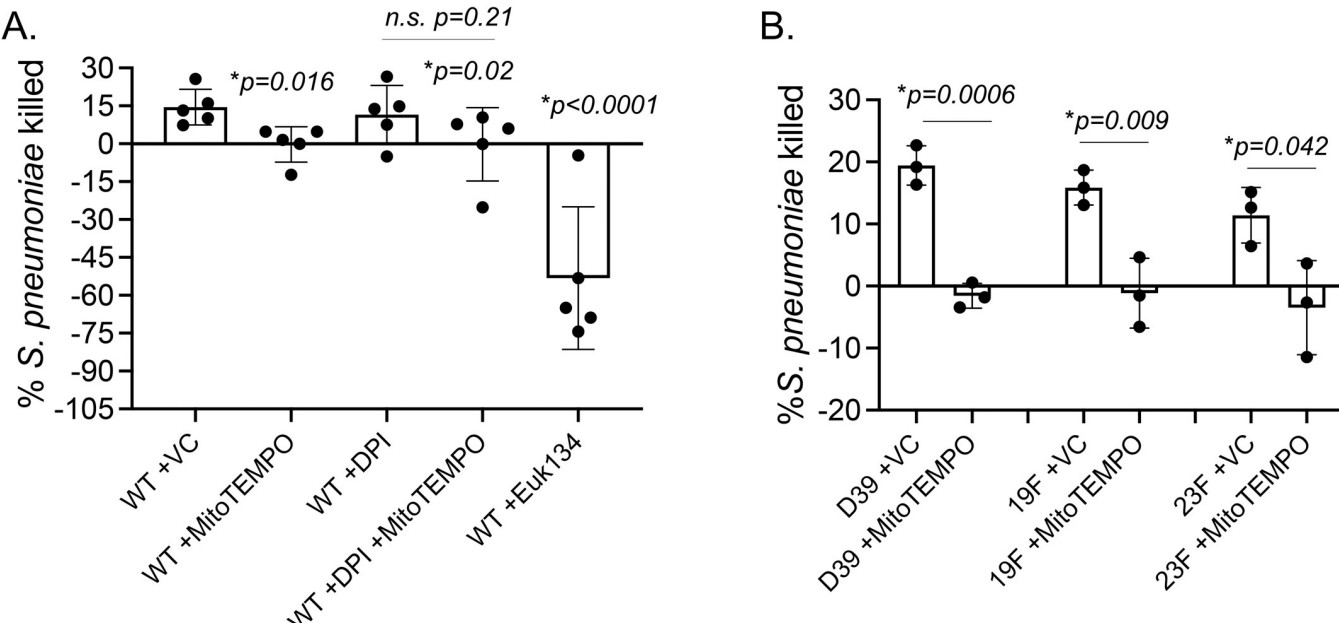

**Fig 4. Mitochondrial ROS are required for the ability of PMNs to kill *S. pneumoniae*.** (A) WT (C57BL/6) bone-marrow derived PMNs were treated with vehicle control (VC), the mitochondrial ROS scavenger MitoTEMPO or Euk134 or the NADPH oxidase inhibitor DPI and then infected with *S. pneumoniae* TIGR4. (B) WT (C57BL/6) bone-marrow derived PMNs were treated with vehicle control (VC) or MitoTEMPO and infected with *S. pneumoniae* D39, 19F, or 23F. The percentage of bacterial killing was determined with respect to no PMN controls under the same treatment conditions. Data are pooled from (A) n = 5 and (B) n = 3 separate experiments. Bar graphs represent the mean +/-SD. * indicates significant differences between the indicated groups as determined by (A) one-way ANOVA followed by Tukey's multiple comparisons test or (B) unpaired Student's t-test. n.s. indicates not significant.

antimicrobial function of PMNs, we treated PMNs with MitoTEMPO and measured the ability of PMNs to kill *S. pneumoniae* TIGR4 strain using an established opsonophagocytic (OPH) killing assay [2,12,32–34, 49]. We found that treatment with MitoTEMPO lead to a complete abrogation of bacterial killing by PMNs (Fig 4A). This was also confirmed using EUK134, another ROS scavenger that can localize to the mitochondria (Fig 4A). We also tested the effect of inhibiting NADPH oxidase activity using DPI, and found that DPI had no significant impact on pneumococcal killing by PMNs (Fig 4A). Further, combining DPI with Mito-TEMPO had no additive effect on PMN antimicrobial activity (Fig 4A). As *S. pneumoniae* strains can vary in their interaction with host cells [50], we also tested the role of MitROS in PMN activity against pneumococcal strains across serotypes 2, 19F and 23F. We found that MitoTEMPO impaired the ability of PMNs to kill all strains of *S. pneumoniae* tested (Fig 4B). These data demonstrate that MitROS are required for the ability of PMNs to kill *S. pneumoniae*.

## Signaling through the A2B adenosine receptor regulates mitochondrial ROS production by PMNs

Next, we wanted to identify host pathways that control MitROS levels in PMNs. We previously found that the extracellular adenosine pathway controls PMN responses during *S. pneumoniae* infection. Signaling via Gs-coupled extracellular adenosine receptors were previously found to inhibit NADPH oxidase activity and ROS production [38–40], which led us to investigate the role of the Gs-coupled A2B adenosine receptor in ROS production by the mitochondria in PMNs. We first asked if A2BR signaling influences intracellular ROS production in PMNs. Using the luminol chemiluminescent-based assay, we compared intracellular ROS production in PMNs from WT C57BL/6 and A2BR[-/-] mice following *in vitro* infection with *S. pneumoniae*.

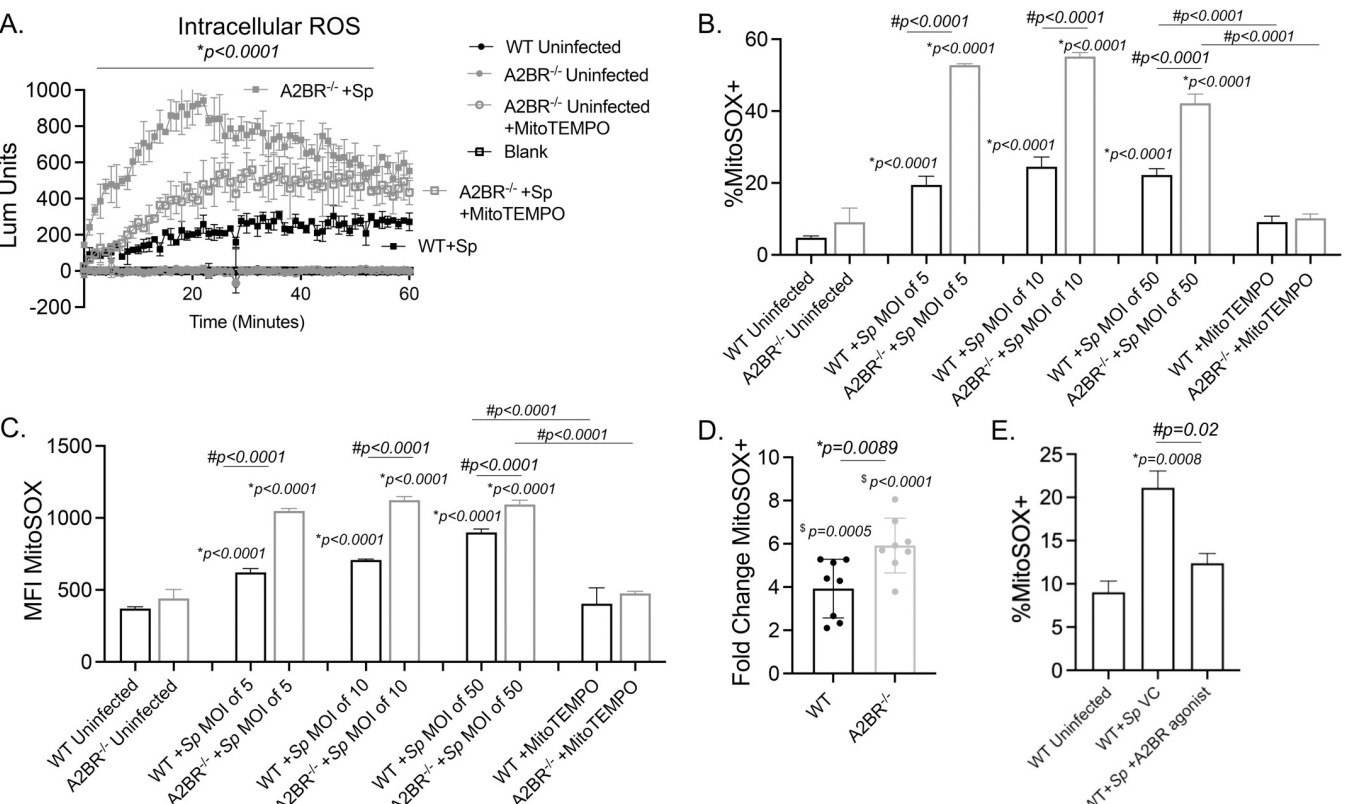

**Fig 5. A2B adenosine receptor signaling blunts mitochondrial ROS production.** (A) A2BR[-/-] (grey) and WT (black) bone-marrow derived PMNs were mock treated (uninfected) or infected with *S. pneumoniae* (+Sp) TIGR4 at a multiplicity of infection (MOI) of 50 in the presence or absence of MitoTEMPO. Cells were monitored for intracellular ROS production over 60 minutes using luminol. Data are representative from 1 of 3 experiments in which n = 3 technical replicates were used per condition. * indicates significant differences between infected A2BR[-/-] and WT controls as well as infected A2BR[-/-] +/- MitoTEMPO treatment as determined by 2-way ANOVA followed by Tukey's multiple comparisons test. Line graphs represent the mean +/-SD. (B-C) WT (Black) and A2BR[-/-] (light grey) marrow-derived PMNs were infected with *S. pneumoniae* TIGR4 at the indicated MOIs for 10 minutes. +MitoTEMPO were treated with the drug prior to infection at an MOI of 50. (B) The % of MitoSOX+ cells as well as (C) the amount of MitoSOX produced (geometric MFI) were determined using flow cytometry. (B-C) Representative data shown are from 1 out of 8 separate experiments in which n = 3 technical replicates were used per condition. Bar graphs represent the mean +/-SD. * indicates significant differences from uninfected controls and # indicates significant differences between the indicated groups as measured by one-way ANOVA followed by Tukey's multiple comparison test. (D) Fold increases in MitoSOX+ cells upon bacterial infection were calculated by dividing the values of infected conditions by uninfected controls for each mouse strain. Data pooled from eight separate experiments (n = 8 mice/group) are shown. $ indicates significantly different from 1 as measured by one-sample t-test and * indicates significant differences between the indicated groups as measured by Student's t-test. (E) WT PMNs were treated with vehicle control (VC) or the A2BR Agonist BAY60-6583 for 30 minutes. Cells were then mock-treated (Uninfected) or infected with *S. pneumoniae* TIGR4 at a MOI of 10 for 10 minutes. The % of mitochondrial ROS producing cells were determined by flow cytometry. Representative data shown are from 1 out of 5 separate experiments in which n = 3 technical replicates were used per condition. Bar graphs represent the mean +/-SD. * indicates significant differences from uninfected controls and # indicates significant differences between the indicated groups as measured by one-way ANOVA followed by Tukey's multiple comparison test.

We found that at baseline neither WT nor A2BR[-/-] PMNs produce ROS and all uninfected conditions were comparable to blank controls (Fig 5A). However, upon infection, intracellular ROS production was detected in both WT and A2BR[-/-] PMNs (Fig 5A). Importantly, the amount of intracellular ROS produced was significantly higher in A2BR[-/-] PMNs compared to WT controls, where A2BR[-/-] PMNs produced 3-fold more ROS at the peak of the response (Fig 5A). This increased response was in part dependent on MitROS, as treatment with Mito-TEMPO reduced intracellular ROS production in A2BR[-/-] PMNs to levels closer to those in WT PMNs (Fig 5A). Further, when we directly measured MitROS production using the flow-cytometry based assay, we found that significantly more A2BR[-/-] PMNs produce mitochondrial ROS than WT PMNs at every MOI tested (Fig 5B). Similarly, the amount of MitROS made (geometric MFI) by PMNs in response to infection was significantly higher in A2BR[-/-]

mice compared to WT controls at every MOI tested (Fig 5C). While both strains of mice upregulated MitROS production in response to infection, this increase was significantly higher in A2BR$^{-/-}$ mice (Fig 5D). As a control, incubation with MitoTEMPO reduced the amount and percentages of MitROS detected to baseline in both strains of mice (Fig 5B and 5C). In contrast, activating A2BR signaling in WT PMNs using a specific A2BR agonist BAY 60–6583 blunted MitROS production in response to *S. pneumoniae* infection (Fig 5E). Given that mitochondrial function is linked to cell death pathways [51], we tested if the increase in mitochondrial ROS production seen in A2BR$^{-/-}$ mouse PMNs was due to an increase in cell death during infection. To test that, we used annexin V and PI staining to measure apoptosis and necrosis of WT and A2BR$^{-/-}$ PMNs infected with *S. pneumoniae*. We found similar levels of apoptotic (PI- Annexin+), necrotic (PI+ Annexin+), and live cells (PI- Annexin-) in PMNs from both mouse strains (S1 Fig) suggesting the differences found in MitROS production were not due to differences in cell viability. In summary, these findings suggest that A2BR signaling blunts MitROS production in *S. pneumoniae* infected PMNs.

## Signaling through the A2B adenosine receptor impairs PMN antimicrobial function

Given our finding that A2BR regulates MitROS production in PMNs, we wanted to determine the functional relevance of this. We first tested the effect of activating A2BR signaling on bacterial killing using the OPH killing assay. We found that treatment of WT PMNs with the A2BR agonist BAY 60–6583 completely inhibited the ability of these cells to kill *S. pneumoniae* TIGR4 (Fig 6). To further confirm that A2BR signaling impairs PMN function, we compared the ability of WT and A2BR$^{-/-}$ PMNs to kill bacteria. We found that A2BR$^{-/-}$ PMNs were significantly more efficient at killing *S. pneumoniae* TIGR4 and displayed a 2-fold increase in bacterial killing (Fig 6). This increase in antimicrobial activity was dependent on MitROS production as treatment with MitoTEMPO reduced killing by A2BR$^{-/-}$ PMNs to WT levels (Fig 6). These data indicate that signaling via the A2B adenosine receptor impairs PMN antimicrobial activity by blunting MitROS responses.

## Signaling through the A2B adenosine receptor impairs host resistance to *S. pneumoniae* infection

We then tested the role of A2BR in host resistance against *S. pneumoniae* infection. To do that, we challenged WT and A2BR$^{-/-}$ mice intra-tracheally (i.t.) with ~5x10$^4$ CFU of *S. pneumoniae* TIGR4, a strain that results in pneumonia that progresses to bacteremia, allowing us to examine pulmonary as well as systemic responses. When we compared bacterial burden, we found no differences in pulmonary bacterial numbers between mouse strains at 6-, 18-, or 48-hours post infection (Fig 7A). In contrast, we did find significantly lower incidence and number of bacteria in the blood of A2BR$^{-/-}$ mice post infection (Fig 7B). This protection from systemic spread translated to better disease outcome where A2BR$^{-/-}$ mice had significantly lower clinical score compared to WT mice (Fig 7C) and significantly enhanced survival following infection (75% vs 25% overall survival) (Fig 7D). These findings demonstrate that A2BR signaling impairs host resistance to *S. pneumoniae* infection.

## Signaling through the A2B adenosine receptor blunts MitROS production by circulating PMNs during *S. pneumoniae* infection

To determine if resistance of A2BR$^{-/-}$ mice against systemic spread was associated with differences in MitROS production, we measured MitROS production in pulmonary and circulating

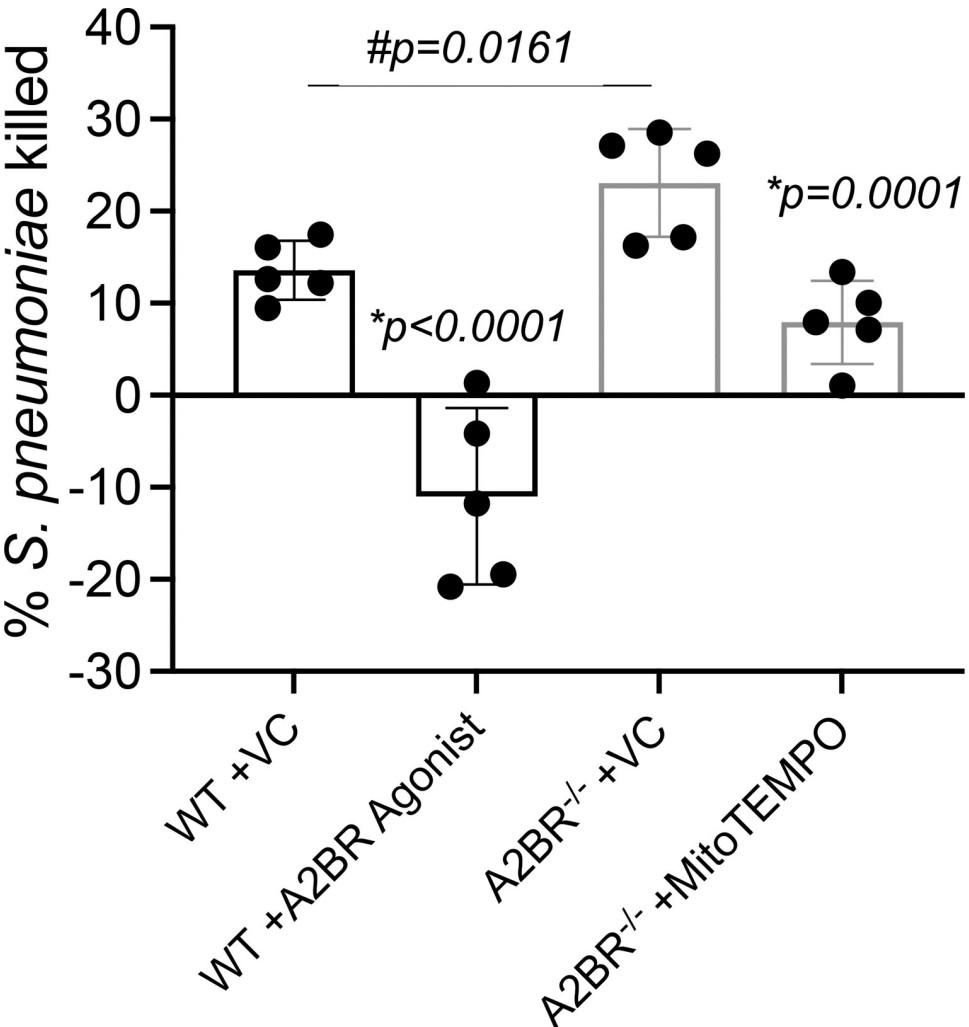

**Fig 6. A2B adenosine receptor signaling impairs PMN antibacterial function.** WT (C57BL/6) (black) or A2BR[-/-] (light grey) bone-marrow derived PMNs were treated with vehicle control (VC), the mitochondrial ROS scavenger MitoTEMPO, or the A2BR agonist BAY 60–6583 and then infected with *S. pneumoniae* TIGR4. The percentage of bacterial killing was determined with respect to no PMN controls under the same treatment conditions. Data are pooled from n = 5 separate experiments. Bar graphs represent the mean +/-SD. * indicates significant differences from VC treated controls for each mouse strain and # indicates significant differences between indicated VC treated WT vs A2BR[-/-] groups as measured by one-way ANOVA followed by Tukey's multiple comparison test.

PMNs following infection using flow cytometry (Fig 8A). We focused on the first 18 hours post infection since we had previously found that is when PMNs are most relevant for controlling bacterial numbers [2]. When we gated on Ly6G[+] PMNs, we found there was a significant increase in the numbers of pulmonary PMNs expressing MitROS in both WT and A2BR[-/-] mice (Fig 8B). When we measured MitROS production in circulating PMNs, we found a 10-fold increase in the amount of MitROS producing PMNs in both strains of mice (Fig 8C). Despite A2BR[-/-] mice having significantly lower bacterial numbers in the circulation (Fig 7B), the amount of PMNs producing MitROS was comparable to those of WT controls.

To test if MitROS production by circulating PMNs during systemic *S. pneumoniae* infection is controlled by A2BR signaling, we challenged mice with ~1x10³ CFU of *S. pneumoniae* TIGR4 strain intra-peritoneally (i.p.) to model systemic disease and measured MitROS

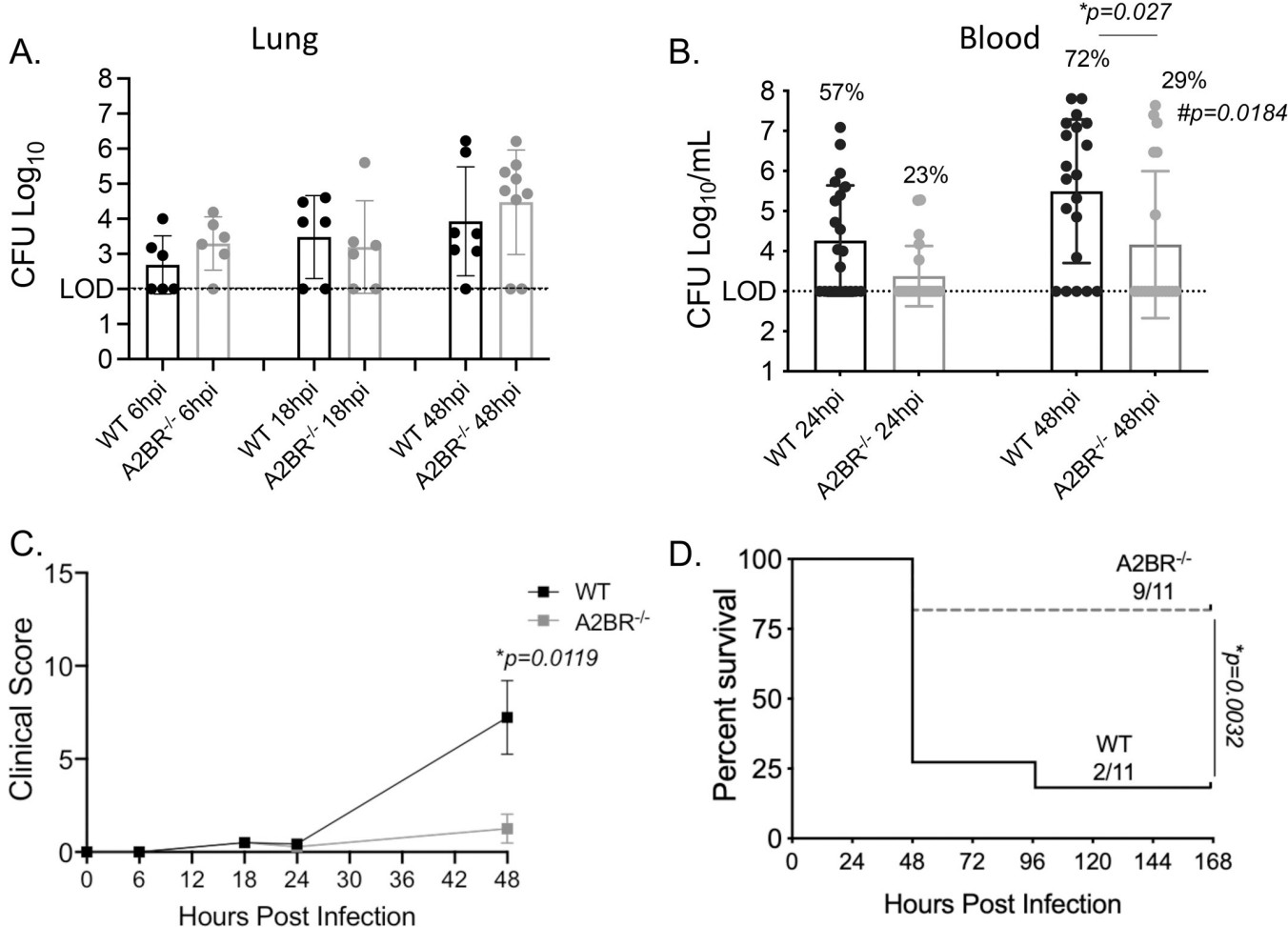

**Fig 7. A2BR impairs host resistance against pulmonary infection with *S. pneumoniae*.** A2BR⁻ᐟ⁻ (light grey) and wild type (WT) C57BL/6 (black) mice were infected with *S. pneumoniae* TIGR4 intra-tracheally. One set of mice were harvested at the indicated hours post infection for enumeration of bacterial numbers in the lungs (A) and blood (B) by plating on blood agar plates. Data are pooled from three separate experiments and each dot represents one individual mouse. * indicates significant differences as measured by unpaired Student's t-test. The percentages indicate the fraction of mice that became bacteremic and # indicates significant differences between the mouse groups in the incidence of bacteremia as measured by Fisher's exact test. (C-D) Another set of mice were monitored for clinical signs and symptoms of diseases (clinical score) (C) and survival (D) over time. Data are pooled from three separate experiments with n = 11 mice per group. * indicates significant differences as measured by Mann-Whitney test (C) and Log-Rank Mantle Cox test (D).

production in circulating PMNs over time. Similar to what we observed during pulmonary infection, we found that i.p. infection resulted in an increase in MitROS producing PMNs in the circulation (Fig 9). By 6 hours post infection, both strains of mice displayed a 2-fold increase in the number of MitROS producing PMNs (Fig 9B) and by 18 hours post infection WT and A2BR⁻ᐟ⁻ mice displayed a 5 and 12-fold increase respectively. The increase in the number of MitROS producing PMNs upon infection was significantly different from uninfected controls only in A2BR⁻ᐟ⁻ mice by 18 hours post infection (Fig 9B). When we measured the amount of MitROS made, by 6 hours post infection, both strains of mice significantly increased the amount of MitROS produced by PMNs compared to baseline. Importantly, the amount of MitROS made by PMNs was significantly higher ($p = 0.0175$) in A2BR⁻ᐟ⁻ mice compared to WT controls (Fig 9C). The increase was consistently maintained in A2BR⁻ᐟ⁻ mice where the amount of MitROS production by PMNs upon infection was again significantly different from uninfected controls only in A2BR⁻ᐟ⁻ mice by 18 hours post infection, while WT

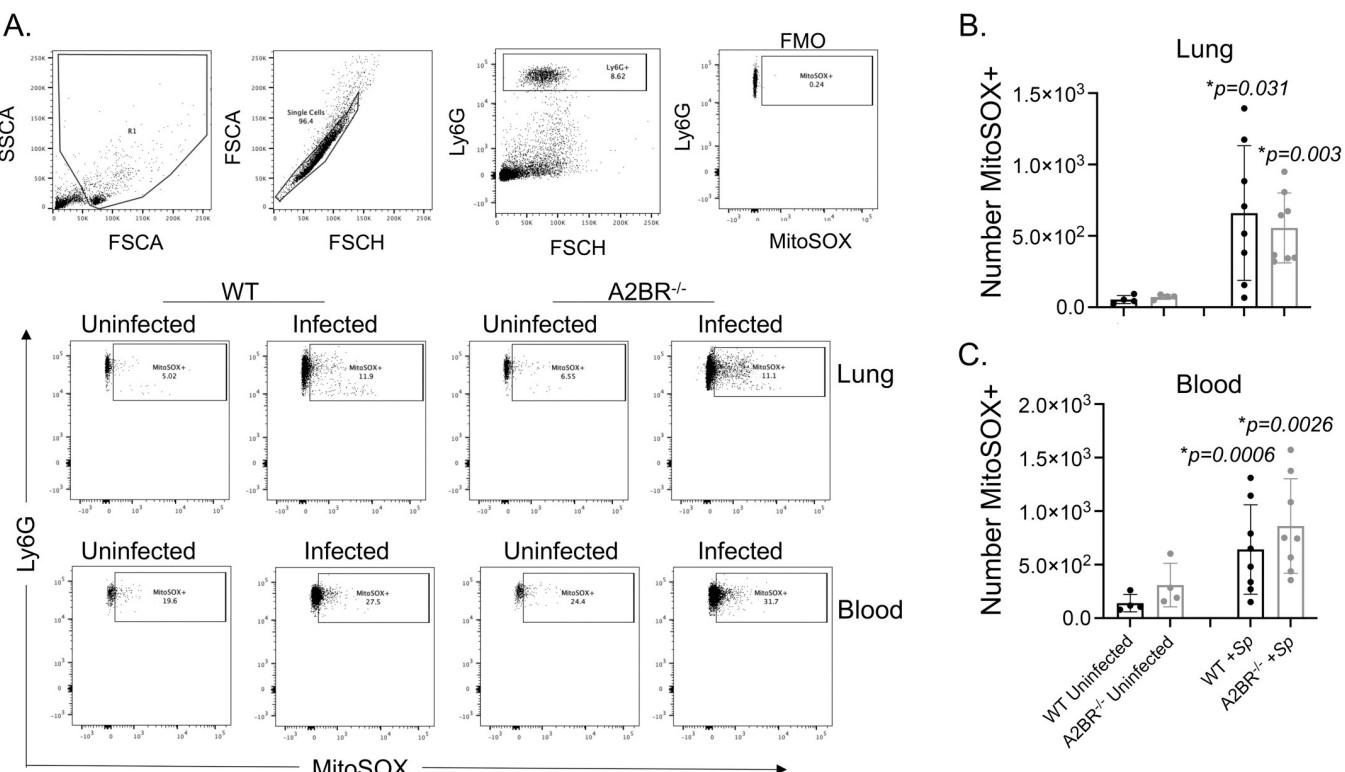

**Fig 8. Mitochondrial ROS are produced by PMNs following pulmonary challenge.** A2BR[-/-] (light grey) mice were infected with *S. pneumoniae* TIGR4 intra-tracheally. At 18 hours following challenge, the number of mitochondrial ROS producing PMNs (Ly6G+) in the lungs (B) and circulation (C) were determined by flow cytometry using MitoSOX (see Materials and Methods). (A) Gating strategy is shown including controls (FMO indicates absence of MitSOX dye only). (B-C) Data are pooled from two separate experiments and each dot represents an individual mouse. (C) * indicates significant differences from uninfected controls for each mouse strain and # indicates significant differences between indicated groups as determined by one-way ANOVA followed by Tukey's multiple comparisons test.

mice displayed variation in their response by that timepoint (Fig 9C). Together, these data suggest that A2BR blunts MitROS production by circulating PMNs during pneumococcal infection.

## Mitochondrial ROS are important for host defense against systemic infection

To assess the role of MitROS in host defense against infection, mice were treated i.p. with the mitochondrial ROS scavenger MitoTEMPO prior to infection. Mice were then challenged i.p. with *S. pneumoniae*, and host survival as well as bacterial burden was monitored over time. As a control for drug toxicity, uninfected controls were treated with MitoTEMPO and no effect on survival of uninfected mice was observed (Fig 10A). However, MitoTEMPO treatment significantly accelerated host death in WT mice challenged with *S. pneumoniae* D39 (Fig 10A). This correlated with an impaired ability to control bacterial numbers systemically, where MitoTEMPO treated mice had a significantly higher bacterial burden in the circulation as compared to vehicle-treated controls (Fig 10B). Similarly, in mice challenged with *S. pneumoniae* TIGR4, none of the MitoTEMPO treated WT controls survived infection, although this was not significantly different from vehicle treated mice (Fig 10C). In *S. pneumoniae* TIGR4 challenged mice, we did not observe a significant difference in early bacterial numbers (Fig 10D), however a higher but not statistically significant fraction of controls cleared the infection

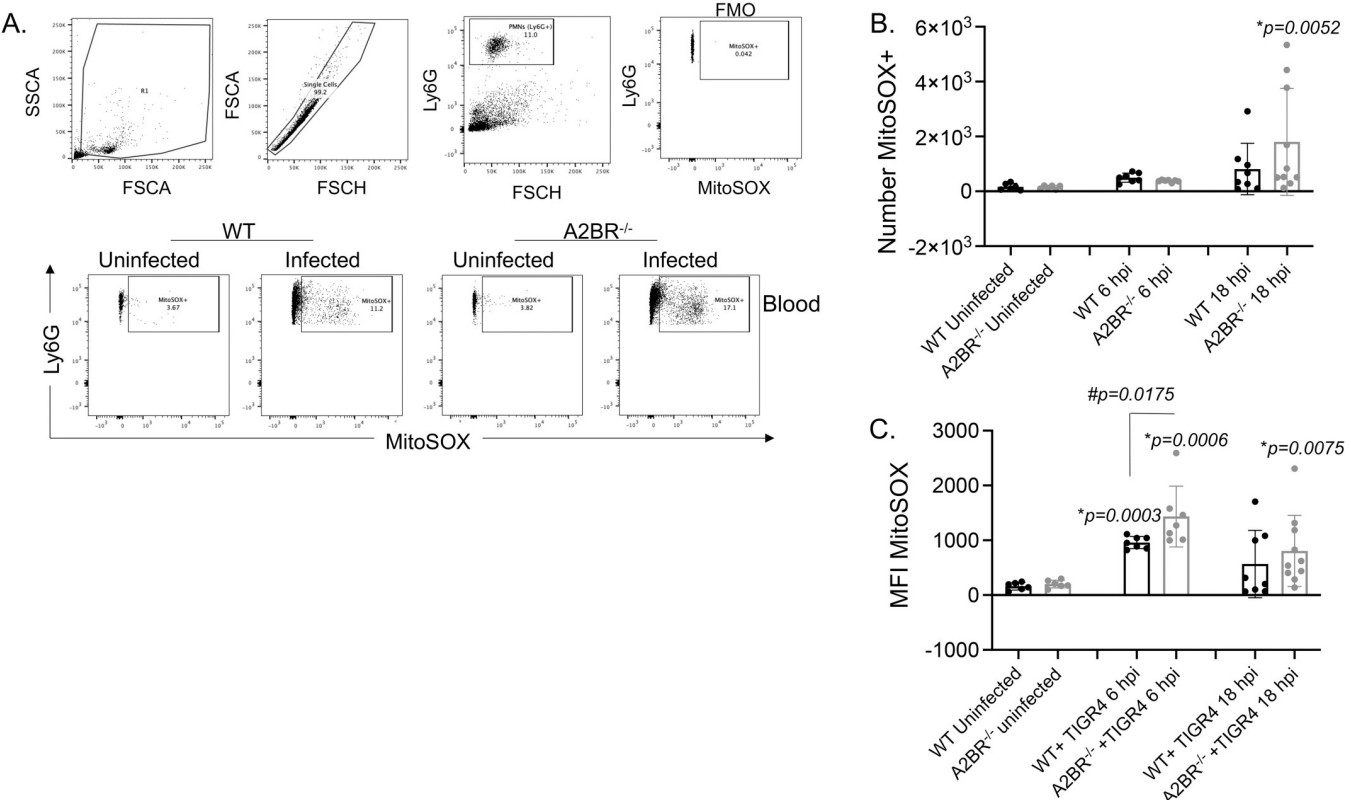

**Fig 9. Mitochondrial ROS are produced in circulating PMNs in response to systemic infection.** A2BR<sup>-/-</sup> (light grey) mice were infected with *S. pneumoniae* TIGR4 intra-peritoneally. At 6 and 18 hours following challenge, the number (B) of mitochondrial ROS producing PMNs (Ly6G+) and (C) the amount of mitochondrial ROS produced (MFI MitoSOX) by PMNs in the circulation were determined by flow cytometry using MitoSOX (see Materials and Methods). (A) Gating strategy is shown including controls (FMO indicates absence of MitSOX dye only). (B-C) Data are pooled from four separate experiments and each dot represents an individual mouse. * indicates significant differences from uninfected controls for each mouse strain as determined by Kruskal-Wallis test.

compared to MitoTEMPO treated mice (~17% vs 0% respectively) (Fig 10E). The effect of MitROS inhibition was much more prominent in D39 vs TIGR4 challenged mice. This could be since WT control mice are extremely susceptible to TIGR4 and thus a further increase in susceptibility is hard to observe. Compared to D39, TIGR4 infected mice got sick faster where death observed before 24h (Fig 10A vs 10C) and mice had a 10-fold higher bacterial burden even at 6 hours post challenge (Fig 10B vs 10D). Overall, these findings suggest that MitROS contribute to host resistance against systemic infection with *S. pneumoniae*.

To determine if the higher MitROS production by circulating PMNs in A2BR<sup>-/-</sup> mice conferred protection against systemic infection, we compared survival across mouse strains. Upon systemic challenge with *S. pneumoniae* TIGR4, A2BR<sup>-/-</sup> mice displayed significantly enhanced survival compared to WT controls (Fig 10C). In comparing bacterial numbers early within the first 18 hours following infection where all mice were still alive, A2BR<sup>-/-</sup> mice displayed a slight decrease in bacterial numbers at 6 hours post infection (Fig 10D), a timepoint where higher amounts of MitROS were made by PMNs in these mice (Fig 9C). Additionally, over time, a significantly higher number of A2BR<sup>-/-</sup> mice cleared the infection compared to WT controls (~60% vs 17% respectively) (Fig 10E). This enhanced protection in A2BR<sup>-/-</sup> mice was lost when mice were treated with MitoTEMPO. MitoTEMPO treatment impaired the ability of A2BR<sup>-/-</sup> to control bacterial numbers early on at 6 hours post infection (Fig 10D), impaired bacterial clearance (Fig 10E) and further significantly accelerated host death where all of the

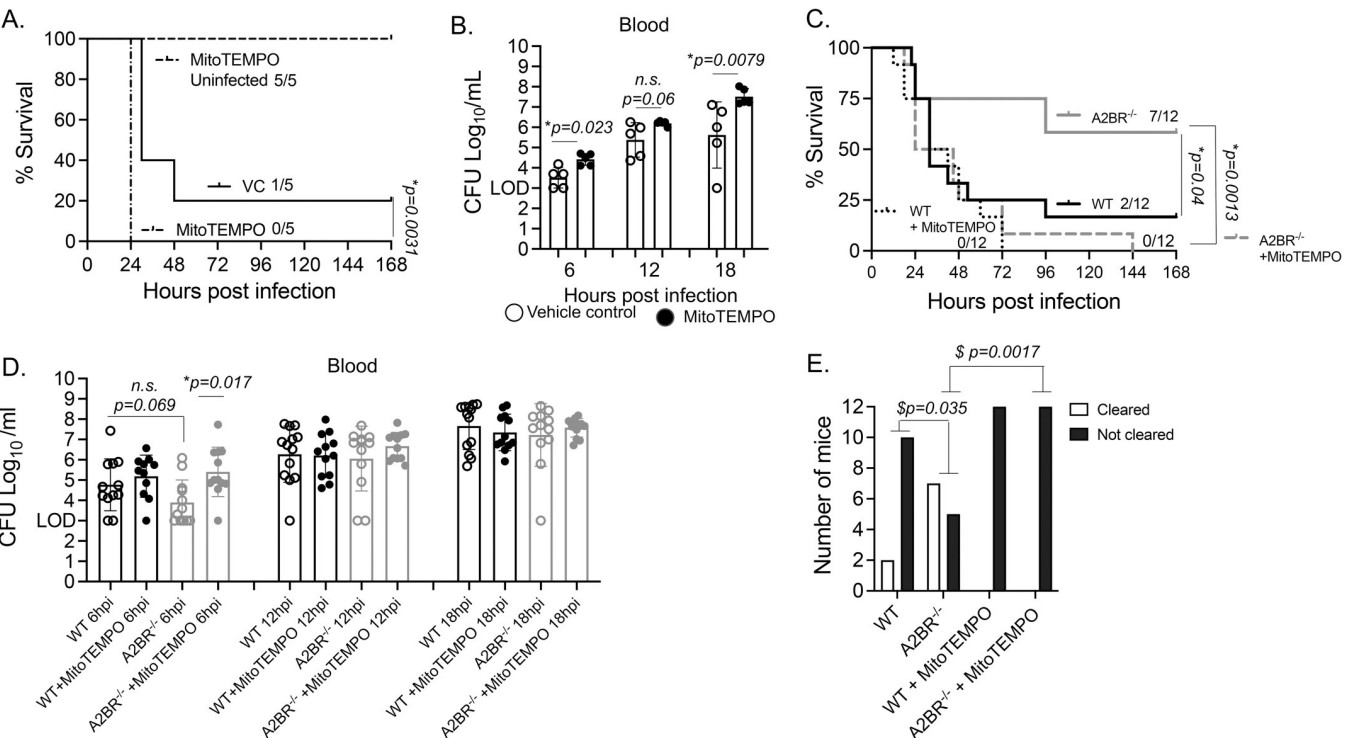

**Fig 10. A2BR impairs host resistance against systemic infection with *S. pneumoniae*.** (A-B) WT C57BL/6 mice treated with vehicle control or the mitochondrial ROS scavenger MitoTEMPO were infected with *S. pneumoniae* D39 intra-peritoneally. (A) survival as well as (B) bacteremia was followed over time. Data shown are from two separate experiments with n = 5 mice per group. * indicates significant differences as measured by Log-Rank Mantle Cox test (A) and Mann-Whitney test (B). (C-D) A2BR[-/-] and WT C57BL/6 mice treated with vehicle control or the mitochondrial ROS scavenger MitoTEMPO were infected with *S. pneumoniae* TIGR4 intra-peritoneally. (C) survival as well as (D) bacteremia and clearance of infection (E) was followed over time. Data shown are from three separate experiments with n = 12 mice per group. * indicates significant differences as measured by Log-Rank Mantle Cox test (C) and Kruskal Wallis test (D). $ indicates significant differences between the mouse groups in the clearance of bacteremia as measured by Fisher's exact test (E). n.s. indicates not significant.

treated A2BR[-/-] mice succumbed by day 3 post infection (Fig 10C). These data demonstrate that MitROS are crucial for enhanced survival of A2BR[-/-] mice against systemic pneumococcal infection. Taken together, the findings further suggest that A2BR signaling impairs host resistance against systemic *S. pneumoniae* infection and that is in part mediated by controlling MitROS production.

## Mitochondrial ROS are required for the antimicrobial activity of human PMNs

Having found that MitROS production in PMNs are regulated by A2BR signaling and are key for host defense against *S. pneumoniae* in mouse models, we sought to determine the clinical relevance of these findings in human PMNs. To do so we first isolated PMNs from the circulation of healthy human donors and measured MitROS production in PMNs using the flow cytometry-based assay. We found that while the magnitude of the response varied between donors as expected, MitROS were produced in a dose dependent manner in response to *S. pneumoniae* infection in all donors tested (Fig 11A). We then tested the role of ROS production in human PMN antimicrobial activity using OPH bacterial killing assays and found that using ascorbic acid, to scavenge out ROS, or MitoTEMPO to scavenge MitROS in particular impaired the ability of PMNs to kill *S. pneumoniae* TIGR4 in all donors tested (Fig 11B). As previously reported [11], DPI which inhibits ROS production by the NADPH oxidase [52], did

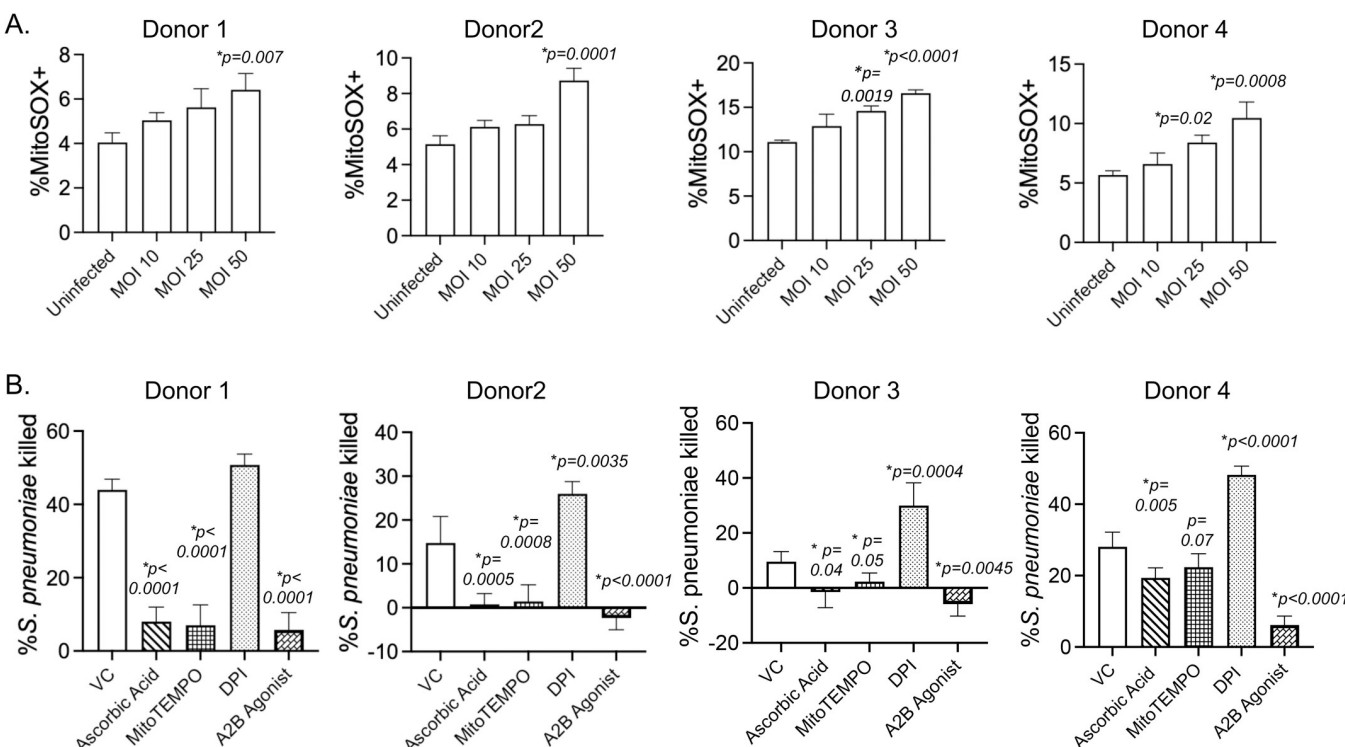

**Fig 11. Mitochondrial ROS are required for the antimicrobial activity of human PMNs.** PMNs were isolated from the blood of young healthy donors and (A) infected with *S. pneumoniae* TIGR4 at the indicated MOIs or mock-treated (uninfected) for 10 minutes. The % of MitoSOX+ cells were determined using flow cytometry. Data shown are from 4 separate donors where each condition was tested in triplicates per donor. Bar graphs represent the mean +/-SD. * indicates significant differences from uninfected controls as determined by one-way ANOVA followed by Dunnett's multiple comparisons test. (B) PMNs were pre-treated with vehicle control (VC), the general ROS scavenger Ascorbic Acid, the mitochondrial ROS scavenger MitoTEMPO, the NADPH oxidase inhibitor Diphenyleneiodonium chloride (DPI) or the A2B Agonist (BAY 60–6583) and then infected with *S. pneumoniae* TIGR4. For each donor, the average percent bacterial killing compared to a no PMN control was calculated from triplicate wells per condition. Data from 4 donors are shown. Bar graphs represent the mean +/-SD. * indicates significant differences from VC treated PMNs as determined by one-way ANOVA followed by Dunnett's multiple comparisons test.

not blunt the ability of PMNs to kill *S. pneumoniae*. These data suggest that mitochondrial ROS produced by human PMNs in response to *S. pneumoniae* are required for efficient PMN antimicrobial function. We finally assessed the role of A2BR and found that activating signaling via this receptor using the specific agonist BAY 60–6583 almost abrogated the ability of PMNs from all donors to kill bacteria (Fig 11B) indicating that that A2BR signaling blunts human PMN antimicrobial function.

## Discussion

ROS production by PMNs was believed to be dispensable for the ability of PMNs to kill *S. pneumoniae* given that the NADPH oxidase is not required for host defense against these bacteria [7–11]. Here we show that the mitochondria act as an alternative source of intracellular ROS and that MitROS are required for both PMN antimicrobial function as well as host defense against *S. pneumoniae* infection. We importantly verify this in PMNs from human donors and show that overall ROS production is required for antimicrobial function of these cells and that the mitochondria, but not the NADPH oxidase are the critical source for antimicrobial ROS. In exploring the host and bacterial factors involved, we found that recognition of bacterial products is sufficient to trigger this response and that signaling via the extracellular A2B adenosine receptor controls MitROS production (Fig 12). This study expands our limited

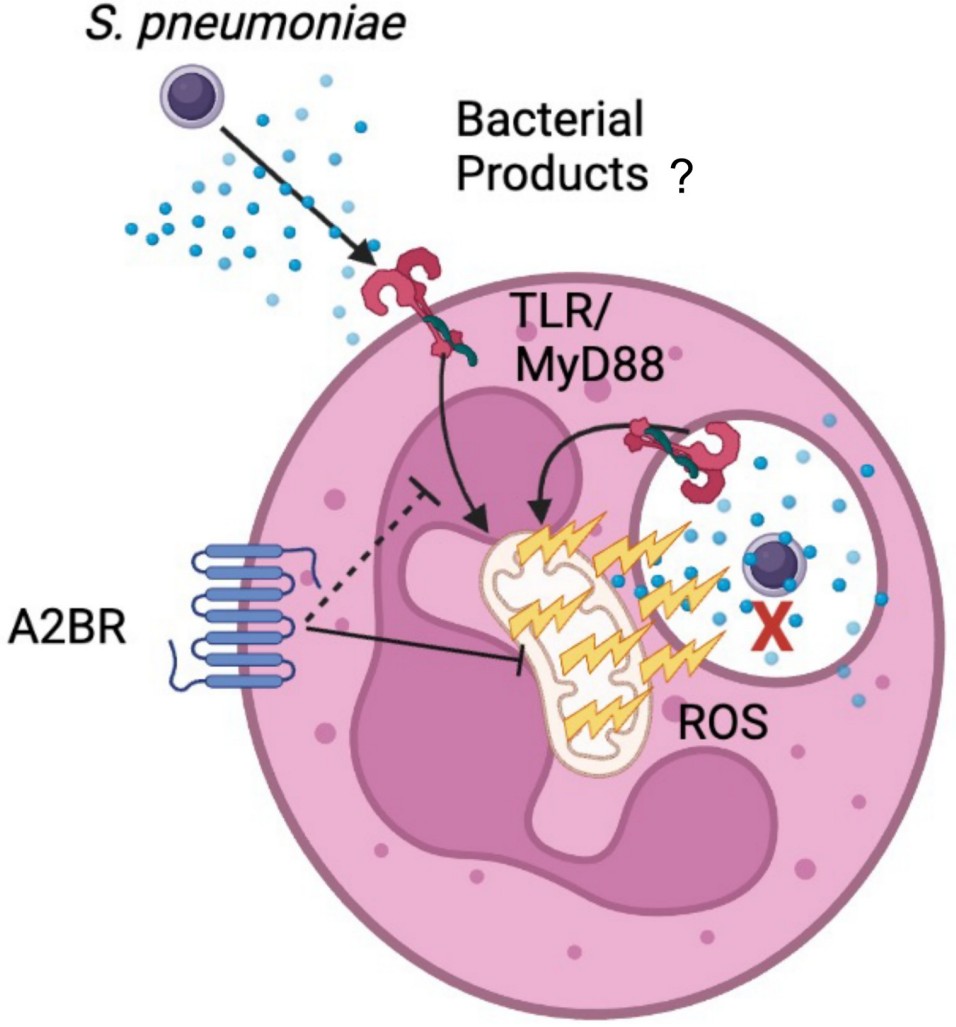

**Fig 12. Model figure.** *S. pneumoniae* induces mitochondrial ROS production by PMNs in a manner partially dependent on MyD88. This response is regulated by A2B receptor signaling and is required for the antimicrobial activity of PMNs. Images were created with Biorender.com.

understanding of MitROS production by PMNs and importantly identifies for the first time a host signaling pathway that negatively regulates this response.

The importance of the mitochondria but not NADPH oxidase as a source of ROS required for control of pneumococcal numbers could be explained in several ways. It is possible that *S. pneumoniae* infection of PMNs triggers ROS production from the mitochondria only. However, that is not likely as previous studies have demonstrated that pneumococcal infection triggers assembly of the NADPH oxidase complex [47] and that inhibition of its activity by DPI blunts overall intracellular ROS levels in response to *S. pneumoniae* infection [52]. Therefore, a more likely explanation is that when the NADPH oxidase is absent or inhibited, that ROS produced by the mitochondria are sufficient to mediate bacterial killing. Our findings support that as we see that ~50% of overall ROS produced by PMNs in response to infection are mitochondrial derived. Alternatively, it is also possible that localization of the assembled NADPH oxidase complex vs mitochondrial derived ROS is the key. In macrophages, it was found that infection with the *E. coli* results in mitochondrial recruitment to the phagosome [18].

Similarly, infection of epithelial cells with respiratory syncytial virus results in nuclear localization of ROS producing mitochondria [53]. On the other hand, infection of macrophages with *S. aureus* results in mitochondrial-derived vesicles containing anti-bacterial hydrogen peroxide that localize with bacterial containing phagosomes [16]. *S. pneumoniae* are extracellular bacteria that are killed very efficiently once engulfed by PMNs [34,49]. However, the dynamics of MitROS production by PMNs and their compartmentalization with the cell remain open questions and will be the focus of future studies in our laboratory.

MitROS can kill bacteria in several ways either directly or by activating other antimicrobial responses. The majority of work exploring this has been done in macrophages and only a handful of studies has been performed in PMNs [15]. In macrophages, MitROS were shown to directly be delivered to the bacteria either by colocalization of the mitochondria or delivery of mitochondrial-derived vesicles with the bacterial-containing phagosomes [16,18]. Additionally, MitROS production in macrophages results in production of proinflammatory cytokines and therefore can indirectly help control bacterial replication by activating and recruiting immune cells [15,19]. In PMNs, MitROS production has been primarily explored using stimuli [22–24,26] and to our knowledge only two studies used pathogens [25,54]. Activation of PMNs by fMLP resulted in release of primary and secondary granules [23] and their activation by fMLP, $Ca^{2+}$ ionophores or antigen/antibody complexes results in increased NETosis which was dependent on MitROS production [23,26]. In one study using live bacteria, infection of PMNs with methicillin resistant *S. aureus* resulted in MitROS production that also triggered NET formation and IL-1β production within four hours of infection [25]. Here, we observed very rapid MitROS production within minutes of infection and at a time where NETs have not had yet time to form [49], and inhibition of MitROS impaired bacterial killing by PMNs within 40 minutes of infection (the duration of the OPH assay). Therefore, it is possible that what we observe here is the direct antimicrobial activity of mitochondrial-derived ROS on bacteria. This is likely to be mediated by superoxide and not hydrogen peroxide as described for *S. aureus* [16] as the dyes we use here directly measure superoxide [23] and we show that infection with Δ*sodA S. pneumoniae*, a bacterial mutant lacking the enzyme superoxide dismutase, a manganese-dependent enzyme that detoxifies superoxide radicals [48], results in significantly increased MitROS levels. We further previously found that Δ*sodA S. pneumoniae* is more susceptible to killing by PMNs and that this susceptibility can be reversed by addition of ROS scavengers [12]. In fact, *S. pneumoniae* have dedicated a significant portion of their genome to ROS detoxification in general [55–59], which highlights the importance of this response for controlling bacterial infection.

In exploring how MitROS are produced, we found that soluble bacterial factors are sufficient to trigger MitROS production by PMNs in a manner partially dependent on MyD88. It is yet to be determined whether pneumococci sensed in the extracellular environment or intracellularly are triggering this response. *S. pneumoniae* can be sensed by surface pathogen recognition receptors [60] including TLR 2 and 4 [61–64], and in macrophages these TLRs have been described to enhance MitROS production or trigger localization of the mitochondria near engulfed bacteria with other pathogens including *E. coli*, *L. monocytogenes* and *Salmonella* [16–21]. Pneumococcal products can also be sensed intracellularly within the cytosol, where it has been shown that the cholesterol-dependent pore-forming toxin PLY allows cytosolic localization of bacterial cell wall components including capsular polysaccharides [65]. Intriguingly, we found here that MitROS production is independent of capsular polysaccharides as well as PLY, despite reports in the literature that PLY can activate the NADPH oxidase [47,52] and can cause direct mitochondrial damage [66]. These data are in line with studies in macrophages showing that MitROS production in response to *L. monocytogenes* infection is also independent of the cholesterol-dependent pore-forming toxin listeriolysin O [20].

Alternatively, it is possible that bacterial metabolites including bacterial-derived ATP or ROS also contribute to this response. In PMNs, MitROS production was described to occur in response to endoplasmic reticulum (ER) stress [25,26]. Infection with *S. aureus*, or TLR7/8 signaling downstream of stimulation with antigen/antibody complexes resulted in activation of the ER stress sensor IRE1-$\alpha$ which was required for subsequent MitROS production [25,26]. These responses were described to occur within four hours of stimulation and here we detect MitROS production more rapidly within 15 minutes of infection. It is also possible that other immune sensing pathways contribute to this response, as PMNs lacking MyD88 were still able to upregulate MitROS production in response to infection. One such pathway may be the NLRP3 inflammasome, which has been linked to MitROS in several models [67–69]. In summary, it is likely that several pathways are involved in MitROS production by PMNs in response to bacterial infection and future studies will focus on pinpointing all the pathways involved and their kinetics during pneumococcal infection.

While several studies have focused on what triggers MitROS production by immune cells, very little is known regarding what pathways control this response [15]. Although MitROS can help control bacterial numbers during lung infection as reported with *Legionella pneumophila* infection [27,70], its production can also be damaging to the host, particularly in the context of pulmonary inflammation as described in acute lung injury models using LPS as well as influenza A virus infection [27–31]. Thus, host responses that control its production would be crucial for survival. In this study, we found that MitROS production by PMNs in response to *S. pneumoniae* infection is regulated by signaling via the A2B adenosine receptor. The absence of A2BR resulted in higher MitROS production by PMNs observed in response to *ex vivo* infection with the same bacterial dose, as well as *in vivo* but only in the systemic circulation. Interestingly, the amount of pulmonary PMNs producing MitROS and the lung bacterial burden was comparable between WT and A2BR$^{-/-}$ mice. The role of A2BR in inhibiting MitROS and impairing host defense in the circulation but not the lung could be due to reduced activation of A2BR in the pulmonary environment. We previously found that circulating PMNs express high levels of A2BR [34], but it is possible this is reduced upon lung entry and/or the higher amount of EAD required to trigger A2BR (discussed below) is only present in the circulation. A2BR is a Gs-protein coupled receptor that activates adenyl-cyclase and increases cAMP production resulting in activation of several downstream signaling pathways [36]. However, the mechanisms by which A2BR controls MitROS and whether it does so by affecting signaling via the MyD88 pathway or is independent of that are yet to be explored.

We found that A2BR signaling was detrimental to host resistance against pneumococcal infection. The role of A2BR in infections is not that well characterized and our findings are similar to what had been reported with *Klebsiella pneumoniae* lung infection [71] and *Clostridium difficile* intestinal infection [72]. This is in contrast to the role of A2BR in acute lung injury models where it was reported to protect the host from excessive inflammation [73,74]. This highlights how pathogens can modulate the host immune response for their benefit.

The detrimental role of A2B in PMN antimicrobial function we describe here is opposite to the role of the Gi-coupled A1 receptor that we previously found to be required for host resistance and for the ability of PMNs to kill *S. pneumoniae* [34,37]. EAD levels in the extracellular environment are low at baseline but can increase more that 10-fold upon tissue damage and the different adenosine receptors have different affinities to EAD and respond in a dose-dependent manner [36]. This suggests that the extracellular adenosine signaling pathway may be a temporal regulator of PMN responses and host defense against pneumococcal infection. While A1 is a high affinity receptor that is activated at very low concentrations of EAD (EC$_{50}$ <0.5μM), A2B is low affinity and is only activated once EAD builds up to more that 30-fold (EC$_{50}$ between 16–64μM [36]), and thus is likely to be triggered later in infection once host

damage has occurred. Pulmonary damage can lead to a loss of organ function and hypoxia and intriguingly, MitROS were reported to stabilize Hypoxia-inducible factor 1-alpha (HIF-1$\alpha$) [75], which is an oxygen-dependent transcriptional activator that is stabilized at low oxygen levels [76]. In turn, HIF-1$\alpha$ activates the transcription of A2BR [77], thus expression and activation of A2BRs could be a negative feedback mechanism that allows the host to balance successful infection control with resolution of pulmonary inflammation.

In summary, we established here the mitochondria as an important source of ROS that is crucial for host defense against *S. pneumoniae* infection. Importantly, we describe here for the first time A2BR signaling as a host pathway that controls MitROS production by PMNs. These findings could have important clinical implications for the future use of drugs targeting adenosine receptor signaling [78] to modulate host resistance against *S. pneumoniae* and other serious infections.

## Materials and methods

### Ethics statement

All work with mice was performed in accordance with the recommendations in the Guide for the Care and Use of Laboratory Animals published by the National Institutes of Health. All procedures were reviewed and approved by the University at Buffalo Institutional Animal Care and Use Committee (IACUC), approval number MIC33018Y. All work with human donors was approved by the University at Buffalo Human investigation Review Board (IRB). Written informed consent was obtained from all donors.

### Bacteria

*Streptococcus pneumoniae* serotypes 4 (TIGR4 strain), 2 (D39 strain), 23F (P833 strain) and 19F (P1084 strain) were grown to (OD$_{650nm}$ of 0.7–0.8) in Todd–Hewitt broth (BD Biosciences) supplemented with Oxyrase and 0.5% yeast extract at 37˚C/5% carbon dioxide. GFP-expressing TIGR4 [34], pneumolysin-deletion mutant ($\Delta PLY$) [79] and superoxide dismutase deletion mutant ($\Delta sodA$) of TIGR4 were constructed as previously described [12] and grown similar to the wildtype TIGR4 strain. Bacterial aliquots were frozen at –80˚C in growth media with 20% glycerol. Prior to use, aliquots were thawed on ice, washed and diluted in phosphate buffered saline to desired numbers. Bacterial numbers were confirmed by plating serial dilutions on tryptic soy agar plates supplemented with 5% sheep blood agar (Hardy Diagnostics). Where indicated, bacteria were killed by incubation with 3% formalin for 2 hours followed by washing 3x in PBS or heating at 65˚C for 2 hours. Loss of viability confirmed by plating on blood agar.

### Mice

Wild Type C57BL/6J (B6), and A2BR$^{-/-}$ on a B6 background were purchased from Jackson laboratories and bred at specific pathogen free facilities at the University at Buffalo Jacobs School of Medicine and Biomedical Sciences animal facility. Wild Type C57BL/10 (B10), MyD88$^{+/-}$, and MyD88$^{-/-}$ on a B10 background were a kind gift from Dr. Jill Kramer and generated as previously described [80]. These mice were bred at specific pathogen free facilities at the University at Buffalo School of Dentistry animal facility. Mice were matched for age and sex and male and female 8-12-week-old mice were used in all experiments.

### Donors

Healthy human donors (25–35 years old, male and female) were recruited and individuals that were taking medications, were pregnant or had chronic or acute infections within the last two

weeks were excluded from the study. All enrolled donors signed approved informed consent forms. All blood draws occurred in the morning between 9 and 10 AM.

## PMN isolation

For mouse PMN isolation, bone marrow cells were isolated from the femurs of mice as previously described [81] and enriched for PMNs using Histopaque 1119 and 1077 density gradient centrifugation. This method yields PMNs at 85–90% purity (Ly6G$^+$ CD11b$^+$) as verified by flow cytometry. For human PMN isolation, whole blood was obtained with acid citrate/dextrose as an anticoagulant. PMNs were isolated using a 2% gelatin sedimentation technique as previously described [82] which yields active PMNs at 87–90% purity (CD16$^+$CD66b$^+$) as verified by flow cytometry. Isolated PMNs were resuspended in Hanks' Balanced Salt Solution (HBSS)/0.1% gelatin without Ca$^{2+}$ and Mg$^{2+}$, and used in subsequent assays.

## Measurement of mitochondrial ROS

PMNs were re-suspended in HBSS (Ca$^{2+}$ and Mg$^{2+}$ free) and acclimated at room temperature for one hour, then spun down and re-suspended in KRP buffer (Phosphate buffered saline with 5mM glucose, 1mM CaCl$_2$ and 1mM MgSO$_4$) and equilibrated at room temperature for 30 minutes. 5x10$^5$ PMNs were then seeded per well in non-tissue culture (TC) treated round-bottom 96 well plates (Grenier Bio-one). Cells were then treated with the indicated drugs (pharmacological inhibitors/ agonists or ROS scavengers) for 15 minutes, then incubated with 5μM MitoSOX-Red fluorescent dye (Invitrogen) for another 15 minutes at 37˚C. PMNs were then washed and mock-treated with 3% sera alone (uninfected) or infected for 15 minutes at 37˚C with the indicated *S. pneumoniae* strain pre-opsonized with 3% sera at the indicated multiplicity of infection (MOI). For mouse assays, sera used were from the matching mice where PMNs were isolated from. For human assays, baby rabbit serum (Pel-Freeze) was used. For assays using supernatants, 5x10$^6$ pre-opsonized bacteria (corresponding to an MOI of 10) were incubated in KRP buffer alone for 15 minutes and, supernatants collected and used to treat PMNs for 15 minutes. For assays testing if contact was required, PMNs and bacteria were separated using 0.4 μm Transwell filters (Corning), where 5x10$^5$ PMNs were seeded on the bottom well of a 24-well non-TC treated plate and bacteria added to the inner chamber of the Transwell at an MOI of 10 and incubated as described above. Mitochondrial ROS production was measured using flow cytometry.

## Measurement of intracellular ROS

Intracellular ROS production was measured as previously described [12]. Briefly, PMNs were re-suspended in HBSS (Ca$^{2+}$ and Mg$^{2+}$ free) and acclimated at room temperature for one hour, then spun down and re-suspended in KRP buffer and equilibrated at room temperature for 30 minutes. 5x10$^5$ PMNs were added per well to 96-well white LUMITRAC plates (Greiner Bio-One). PMNs were mock-treated with 3% sera alone (uninfected) or infected with *S. pneumoniae* pre-opsonized with 3% matching sera from the same mice. Phorbol 12-myristate 13-acetate (PMA) (Sigma) (100nM) was used as a positive control. Intracellular ROS was detected by addition of 50μM Luminol (Sigma) [46,83–85] and luminescence immediately read (post infection) over one hour at 37˚C in a pre-warmed Biotek Plate reader. Wells with buffer or Luminol alone were used as blanks.

## Detection of apoptosis

Following equilibration in KRP buffer as outlined above, PMNs were mock-treated with 3% sera alone (uninfected) or infected with *S. pneumoniae* pre-opsonized with 3% matching sera

from the same mice for 15 minutes at 37˚C. The percentage of apoptotic cells were then determined using the FITC Annexin V apoptosis detection kit with PI (BioLegend) and flow cytometry as per manufacturer's instructions.

## Opsonophagocytic killing assay (OPH)

The ability of PMNs to kill *S. pneumoniae* was assessed *ex vivo* as previously described [34,82]. Briefly, 100µl reactions of $1x10^5$ PMNs in Hank's buffer/0.1% gelatin were incubated with $1x10^3$ *S. pneumoniae* pre-opsonized with 3% sera (matching sera for mice or baby rabbit sera for human). The reactions were incubated rotating for 40 minutes at 37˚C. Where indicated, PMNs were incubated with the specific drugs (pharmacological inhibitors/ agonists or ROS scavengers) for 30 minutes prior to infection. Percent killing relative to parallel incubations without PMNs under the exact same conditions (+/- treatments) was determined by plating serial dilutions on blood agar plates.

## Mouse infections

For lung infections, mice were intratracheally (i.t.) challenged with *S. pneumoniae* TIGR4 at $5x10^4$ - $1x10^5$ CFU as previously described [34,86]. Following infection, mice were monitored daily over 7 days for survival. Mice were also monitored for clinical signs of disease using the following criteria: weight loss (scored 0 for <5%, 1 for 5–10%, 2 for 10–15% and 3 for >15%), activity level (scored 0 for normal, 1 for slightly diminished, 2 for diminished, 3 for lethargic and 4 for moribund), posture (scored 0 for normal, 1 slight hunch, 2 severe hunch), eyes (scored 0 for normal, 1 each for protruding, sunken, discharge and closed) and breathing (scored 0 for normal, 1 for irregular, 2 for gasping). Scores were added and totaled from healthy [0] to severely sick [15], as previously described [34,86]). Bacteremia was determined for up to 48 hours post infection. At 48 hours post infection mice were euthanized to assess bacterial burden in the lung. Lung homogenates and blood were assessed for CFU by plating on blood agar plates. The limit of detection was $10^2$ CFU per lung and $10^3$ CFU per mL blood. When no colonies were detected on the plates, the numbers of bacteria were noted at the limit of detection (2.0 $\text{Log}_{10}$ in lungs and 3.0 $\text{Log}_{10}$/ mL blood). The effect of Mitochondrial ROS production on host resistance against systemic infection was determined using the mitochondrial ROS scavenger MitoTempo (Cayman Chemical). The mice were given intraperitoneal (i.p.) injections of MitoTempo or vehicle control at days -1, 0 (immediately before challenge), 1, 2 and 3 relative to infection. Mice were challenged i.p. with $5x10^2$ - $1x10^3$ CFU *S. pneumoniae* TIGR4 or D39 as indicated. Host survival, systemic bacterial burden and clinical score was followed over time.

## Measurement of mitochondrial ROS *in vivo*

For examining ROS production *in vivo* following bacterial challenge, mice were euthanized at the indicated timepoints. Blood (300µl per mouse) was first collected by portal vein snips using acid citrate/dextrose as an anticoagulant and the mice were then perfused with 10mL PBS and the lungs harvested. The lungs were then minced into pieces and digested for 1 hour with RPMI 1640 supplemented with 10% FBS, 1 mg/mL Type II collagenase (Worthington), and 50 U/mL Deoxyribonuclease I (Worthington) at 37˚C/ 5% $CO_2$. Single-cell suspensions were obtained by mashing the digested lungs. Red blood cells were removed by treatment with a hypotonic lysis buffer (Lonza). The cells were resuspended in FACS buffer (HBSS/ 1% FBS), treated with Fc block and surface stained with for Ly6G for 20 minutes at room temperature. For detection of mitochondrial ROS, 5µM of MitoSOX-Red (Invitrogen) was then added and incubated with the cells 20 minutes at 37˚C. Cells were then washed and resuspended in FACS buffer and ROS production measured using flow cytometry.

## Flow cytometry

The following antibodies were purchased from BD Biosciences or Invitrogen and used: Fc block (anti-mouse clone 2.4G2 and anti-human clone 3G8), Ly6G (anti-mouse, clone IA8), CD16 (anti-human clone 3G8) and CD66b (anti-human clone G10F5). Fluorescence intensities were measured on a BD Fortessa. At least 50,000 events for lung tissue, 10,000 events for blood and 5,000 events for *in vitro* purified PMN assays were analyzed using FlowJo.

## Inhibitors, agonists and scavengers used

For *in vitro* assays, the effect of A2BR was assessed using A2BR agonist BAY 60–6583 (Tocris) used at 285.7nM, a concentration corresponding to the 2xKi for the receptor. For *in vitro* assays, the effect of ROS was investigated using the general ROS scavenger Ascorbic Acid (Fisher) at 100μM dissolved in PBS, the NADPH oxidase inhibitor Diphenyleneiodonium chloride (DPI) (Sigma-Aldrich) at 10μM or the mitochondrial ROS scavenger MitoTEMPO at 100μM. For *in vivo* experiments, MitoTEMPO was injected i.p. at 20mg/kg at days -1 to day 3 post infection. All chemicals were dissolved in DMSO (unless indicated) and filter sterilized by passing through a 0.22μm filter. Control groups were treated with diluted DMSO or PBS as appropriate as a vehicle controls.

## Statistical analysis

All statistical analysis was performed using Prism9 (Graph Pad). With the exception of ROS measurements, data shown are pooled from separate experiments. For ROS measurements via flow cytometry and plate-based assays, due to inter-assay variation, representative data are shown, where each condition is tested in triplicates. Bacterial numbers in blood and lungs were log-transformed. Distribution of the data was tested using the D'Agostino & Pearson and Shapiro-Wilk tests. Bar and line graphs represent the mean values +/− standard deviation (SD). For normally distributed data, significant differences were determined with one-way ANOVA followed by Tukey's or Dunnett's multiple comparisons tests or Student's t-test as indicated. For data that failed the normality test, significant differences were determined with Kruskal-Wallis test or Mann-Whitney test as indicated. For incidence and clearance of bacteremia, significant differences were measured by Fisher's exact test. For analysis of survival curves, Log-rank (Mantel-Cox) test was performed. All $p$ values < 0.05 were considered significant.

## Supporting information

**S1 Fig. A2BR does not affect PMN viability.** A2BR[-/-] (light grey) and WT (black) PMNs were mock treated (uninfected) or infected with *S. pneumoniae* TIGR4 at the indicated MOIs and then stained with Annexin-V and PI. (A) Gating strategy. The % of (B) live (Annexin-, PI-), (C) apoptotic (Annexin+, PI-), or (D) necrotic (Annexin+, PI+) cells was determined by flow-cytometry. Bar graphs represent the mean +/-SD. Data are pooled from three separate experiments.
(TIF)

## Acknowledgments

We would like to acknowledge Andrew Camilli, John Leong and Sara Roggensack for bacterial strains. We would like to thank Shaunna Simmons and Alex Lenhard for their critical feedback on the manuscript.

## Author Contributions

**Conceptualization:** Jill M. Kramer, Elsa N. Bou Ghanem.

**Data curation:** Sydney E. Herring, Sovathiro Mao, Manmeet Bhalla, Essi Y. I. Tchalla.

**Formal analysis:** Sydney E. Herring, Sovathiro Mao, Manmeet Bhalla, Essi Y. I. Tchalla, Elsa N. Bou Ghanem.

**Funding acquisition:** Elsa N. Bou Ghanem.

**Investigation:** Sydney E. Herring.

**Project administration:** Elsa N. Bou Ghanem.

**Resources:** Jill M. Kramer.

**Supervision:** Elsa N. Bou Ghanem.

**Writing – original draft:** Sydney E. Herring.

**Writing – review & editing:** Elsa N. Bou Ghanem.

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
