## [Decision Letter · Decision Letter 0]

28 Jul 2022

Dear Dr. Bou Ghanem,

Thank you very much for submitting your manuscript "Mitochondrial ROS production by neutrophils is required for host antimicrobial function against Streptococcus pneumoniae and is controlled by A2B adenosine receptor signaling" for consideration at PLOS Pathogens. As with all papers reviewed by the journal, your manuscript was reviewed by members of the editorial board and by independent reviewers. In light of the reviews (below this email), we would like to invite the resubmission of a significantly-revised version that takes into account the reviewers' comments.

It is important that the authors experimentally address concerns regarding Fig 11, as suggested by Reviewer 2 and noted by Reviewer 1, which are important for the demonstration of the connection in mechanism between A2BR and MitROS in vivo.

We cannot make any decision about publication until we have seen the revised manuscript and your response to the reviewers' comments. Your revised manuscript is also likely to be sent to reviewers for further evaluation.

Sincerely,

Carlos Javier Orihuela, PhD

Associate Editor

PLOS Pathogens

Michael Wessels

Section Editor

PLOS Pathogens

Kasturi Haldar

Editor-in-Chief

PLOS Pathogens

orcid.org/0000-0001-5065-158X

Michael Malim

Editor-in-Chief

PLOS Pathogens

orcid.org/0000-0002-7699-2064

It is important that the authors experimentally address concerns regarding Fig 11, as suggested by Reviewer 2 and noted by Reviewer 1, which are important for the demonstration of the connection in mechanism between A2BR and MitROS in vivo.

Reviewer's Responses to Questions

**Part I - Summary**

Reviewer #1: This study investigates the role of mitochondrial ROS production from neutrophils in defense against Streptococcus pnuemoniae. This is a novel study in an important area of research, represents a significant amount of effort and the experimental data is of high quality. The authors demonstrate clearly that PMN increased the amount of mitochondrial ROS in response to a soluble Spn ligand present in culture supernatant from live cultures, which does not require direct contact (shown clearly with transwell assays). The effect is diminished in the absence of MyD88, which may point towards a role for TLR signaling, though the specific molecule responsible is not investigated further. As identification of the bacterial signal was not the goal of these studies, I do not think additional experiments are needed in this area, however I do think the text should be amended – currently the manuscript (primarily the discussion, but also figure 13) implies that a bacterial ligand is responsible, alluding to pattern recognition receptors or other PAMP motifs. It is possible that a secondary metabolite, bacterial ATP, an unknown cytolytic toxic (other than PLY, which the authors show is not responsible), pH, etc is actually triggering the response instead of a canonical PAMP, though none of these were tested. The authors clearly show a role for MitoROS within PMN for killing in vitro, and I have almost no comments on Figure 1 – 6, I thought this part of the paper was extremely convincing and beautifully done.

The second part of the paper (Figures 7-13) investigates the contribution of A2BR, an adenosine receptor, and seeks to connect A2BR to MitoROS production and resistance to Spn in vivo. While these were another set of nicely conducted studies when analyzed independently, I found the connection between A2BR and MitoROS a bit harder to follow. First, the transition to A2BR was rather abrupt in the text – it was not clear why A2BR was chosen (why did the authors not also investigate A1, A2A, A3?). Further, to my knowledge, adenosine signaling results following ATP conversion to ADP, AMP, and finally adenosine by ectonucleases, and there are a variety of purinergic receptors that would therefore be activated (potentially) before A2BR. This would be unreasonable in scope to address experimentally – but the authors should at least discuss the current knowledge in the text. The authors might consider including stimulations with adenosine (perhaps in combination with inhibitors of the other adenosine receptors) if they think this is a primary mechanism of inducing MitoROS.

The authors provide convincing data that A2BR-/- mice are more resistant to an intratracheal infection model (but not systemic, which is interesting but not discussed in the paper) and show that A2BR-/- neutrophils produce more mitochondrial ROS. I was not convinced by the data as presented that A2BR has a major role in induction of mitochondrial ROS. The authors state that in some conditions, MitoROS in PMN is higher from A2BR-/- mice, and imply this is important in response to Spn, however there is quite a bit of variability in the % of positive cells – is this effect significant in uninfected cells (7B, 10C?). The data representation and statistics make it difficult to compare between figures at times (are uninfected data in 10C and 11B comparable?). Further, its not clear if the MitoROS correlates with CFU, and while MitoTEMPO was used successfully to reverse most of the in vitro studies, they did not rescue the phenotype of mouse survival (though this was only tested in systemic infection, where there is no difference between A2BR and WT mice). There are clearly a number of explanations for this that would may be better addressed in future studies, however this made it difficult to reconcile – as it stands, I think the data is clear that Spn induces MitoROS, and A2BR-/- mice have differential ROS profiles, but am not clear those two results are related, and wonder if this might be better as two publications – the first describing MitoROS and then a followup delving into the details of adenosine signaling with more details. Alternatively, the authors may be able to address this disconnect with textual edits and or minor additional experiments demonstrating a more direct connection between the pathways.

I thought the human PMN data was convincing and nicely explained in context of the field with ROS and DPI vs MitoROS.

Reviewer #2: This is a well-written study which provides novel insights regarding the role of mitochondrial-derived reactive oxygen species (MitROS) in neutrophil-mediated protection against S. pneumoniae. In particular, the demonstration that MitROS contributes to S. pneumoniae killing represents a significant advance by clarifying how despite the fact that the inhibition of NADPH oxidase-dependent ROS has no impact on neutrophil killing (shown by several groups), NADPH oxidase-independent ROS in neutrophils still has a major role in the host defense against S. pneumoniae. In addition, the mechanisms regulating MitROS in neutrophils, versus macrophages, are understudied. Here, the authors identify the adenosine receptor A2B on neutrophils as critical for regulating this response by suppressing MitROS, leading to reduced neutrophil killing and impaired bacterial clearance. This study would be further improved by addressing a few specific comments detailed further below, including adding statistical analysis and/or repeats for a critical in vivo Figure, connecting the pulmonary versus systemic infection model phenotypes more clearly, and addressing whether bacterial-derived ROS contributes to or impacts these findings.

**Part II – Major Issues: Key Experiments Required for Acceptance**

Reviewer #1: The majority of figures should be updated to show data as dot plots instead of bar graphs, and the statistical tests should be checked (T tests require normal distribution, which do not appear to be appropriate in many cases).

Reviewer #2: -Statistics are not shown for Fig 11C to back up conclusion that ‘MitoTEMPO treatment accelerated host death in both strains of mice’, lines 310-311. This is an important point, because Fig 11C provides the only data in vivo demonstrating that the mitochondrial scavenging impacts infection outcomes. While a phenotype is shown with the A2BR-/- mice, the ROS and killing for neutrophils from these mice is only partially dependent on MitROS, so it is important to show that the A2BR and MitROS phenotypes are linked in vivo. It also appears the data in Fig 11C are from one experiment, n=5 mice per group. In this case, the experiment should be repeated to confirm this important finding.

-It is difficult to connect conclusions from the two different infection model systems. For example, what is the interpretation regarding no difference in percent survival for WT vs A2BR-/- mice in the i.p. model (Fig 11), compared to the pulmonary infection where there was a large difference (Fig 9)? To address this concern, it would be helpful to compare blood CFUs for the i.p. model in WT vs A2BR-/- mice, because for the pulmonary model it was shown that A2BR-/- mice have reduced CFUs in the blood but a comparable number of circulating blood PMNs producing MitROS (MitoSOX+). For the i.p. model, there are an increased number of circulating blood PMNs producing MitROS in the A2BR-/- mice but no difference in survival.

-S. pneumoniae itself produces ROS including hydrogen peroxide, and live bacteria (or supernatants) are required for MitROS induction (Fig 2). The potential role of bacterial-derived ROS is not addressed, and it is unclear whether this may influence some of the results presented. For example, can MitoSOX detect bacterial ROS? Or, alternatively, is there ROS in the bacterial supernatants and could this contribute to triggering MitROS?

**Part III – Minor Issues: Editorial and Data Presentation Modifications**

Reviewer #1: Line 300 – “MitROS production by circulating PMN may be important for controlling systemic spread during S. pneumoniae infection.” MitoROS is similar in both strains in both lung and spleen, and does not correlate with CFU. It is equally likely that MitoROS from PMN has no role in controlling Spn given this data. The authors should adjust this conclusion appropriately. Relatedly, does the level of Mitosox + PMN (in Fig 10C) relate to the CFU of the mice (figure 9B?). i.e. do mice with higher Mitosox levels tend to have more or less CFU?

Line 307 / Fig. 11B – “This increase upon infection was significantly different from uninfected controls only in A2BR-/- mice (Fig. 11B).” – This is surprising to me based on the representative plots shown in Fig 11A. The frequency of MitoSOX+ PMN seems to also increase significantly in the WT (3% to 11% is a bigger fold change than seen in figure 10). Please report the frequency in addition the number of cells and perform statistical testing. Also, what were the CFU in the i.p. infected mice (figure 11?)

Line 311 / Fig. 11C – There is no difference in survival between WT and A2BR mice with i.p. infection, so this is a hard experiment to reconcile with the rest of the paper. Looking at the WT alone, with or without MitoTempo, there appears to be no difference in survival, suggesting the mitochondrial ROS is not a main driver of outcome in a normal situation. In contrast, the ABR2 mice with or without MitoTempo looks like it might be significant, but given the spread of the data it is hard to conclude. Do the authors think that theres no role at all, or that there may be a role only in an A2BR-/- mouse (and if so, what would be the reason why?)

There is a difference in survival for i.t. (fig 11 vs fig 9), but not for i.p., - what are the implications for your results? Is it not worth trying the mitoTEMPO in the lung infection model?

Line 312 – is the MitoTempo survival significantly different from the sham treated mice? If not, this data does not support the statement “MitROS are required for host resistance to S. pneumoniae systemic infection.” If they are significant, how do the authors rule out the possibility of a toxic effect of MitoTempo?

Line 317 – “Having established that MitROS production in PMNs are regulated by A2BR signaling” – I think this statement if too bold. The authors show that A2BR-/- leads to increased frequency of MitoSox+ PMN in vitro following stimulation (Fig 7) and that overall, increased levels of ROS (7A) – but I’m not convinced from this manuscript alone that MitoROS is regulated primarily by A2BR.

Line 343 – 344: “we found that recognition of bacterial products is sufficient to trigger this response and that signaling via the extracellular A2B adenosine receptor controls MitROS production”. The clarity of this manuscript would be improved if this part of the discussion briefly summarized the results with a bit more precision, since this sentence builds on results from almost 12 different figures. For example, perhaps something like: we demonstrate that a soluble factor present in culture supernatant is sufficient to induce mitochondrial ROS from PMN… Given that the MyD88-/- cells have reduced, but not abolished MitoSox, and as a specific ligand or signaling pathway was not identified, I think this sentence is a bit strong for the data.

Lines 347-363: Its also possible that in the absence of ROS, MyD88, A2BR etc… is there a differential activation of antimicrobial effectors that lead to differential killing that is independent of ROS? NETosis, degranulation, lysosomal trafficking, autophagy? *note, I see this is addressed later in the discussion (line 373), this should be amended so the writing doesn’t sound as absolute, given the authors are clearly considering the other mechanisms.

Line 375: fmlp should be fMLP

Line 392 onwards: (when discussing the bacterial products and signaling) – is it possible that ATP or adenosine is being released from Spn (for example, following autolysis) that is activating purinergic signaling?

Line 441 onwards: this appears to be the first time the authors discuss other adenosine receptors. A1 only discussed briefly, while A2A and A3 are not discussed. This should be introduced and explained earlier in the paper (why A2B and not the others) and the discussion would be improved by connecting these new data to their previous work. I am not clear what the authors think the role of each of these receptors are. Further, it may be worth summarizing briefly purinergic signaling in general (can adenosine be released directly, or do the authors assume the pathways is following ATP release and subsequent degradation by ectonucleases?).

METHODS

Line 473 – OD 0.7-0.8 is not mid log (perhaps late log or early stationary). How much autolysis is occurring here?

Line 481 – incubation with X% formalin. Was this quenched before adding to cells?

Line 551 – mice were infected with 10-5x104 CFU? Perhaps the authors mean between 1 x104 and 5 x104?

Line 557 (and elsewhere) – ml should be mL (uppercase L, as this is the SI symbol)

Line 563 – dose 1-0.5x104 – what does this mean? Please describe how you determined clincal score

Line 598 – “due to inter-assay variation, representative data are shown”. I understand the challenges with variability but for this reason think the mean from each experiment should be shown. If the raw values are wildly different, perhaps a normalization to media or uninfected control may be useful?

Line 600 – “log transformed to normalize distribution” – log transformation helps with weighting of the average (prevents skewing) but shouldn’t normalize the distribution of the data. Is this what you mean?

Comments on Figures not addressed in text.

Fig 6A – the axis break makes this hard to read. I think it would be easier with a single axis going down to -100

Fig 7A – this is quite hard to read with so many lines and all black / white. Perhaps moving all the uninfected / control to a second graph would make it less confusing (since there are only really 3 lines that are of interest here? Also, it is not clear what the p<0.0001 means as the line goes across all samples. The legend says this is total ROS. Is this why the MitoTempo doesn’t fully block the signal in the A2BR-/- Spn cells? If so, would like to see this with MitoSox as well.

Fig 7B and C – Legend: “Representative data shown are from 1 out of 5 separate experiments in which n=3 technical replicates were used per condition.” For 7B and 7C, please include the results from the 5 replicates (ideally with a dot plot and SEM) – was there significant variability in the experiments?

Figure 8A – please show individual replicates. Significant difference annotation is confusing, does the # p =0.0161 “# indicates significant differences between indicated groups” mean that all three are somehow different, or this is just comparing to WT + VC as well (if so, why this is annotation different from the others?)

Figure 9A and 9B – T tests assume a normally distributed sample, is that the case for these?

Figure 9C – Assume this is clinical score of pulmonary pathology? Please clarify in figure legend and include scoring criteria. Representative images would be a nice addition as well.

Reviewer #2: -In general, the Figures seem unnecessarily spread out. For example, single panel figures Fig 2, Fig 4, and Fig 5 could be combined, as all address the bacterial components which are responsible for inducing MitROS

-It is interesting that the MyD88-/- PMNs still had a significant increase in MitROS in response to Sp, though this was not as robust as that induced in WT PMNs (Fig 5). This suggests that MyD88-independent signals also contribute to some degree. This nuance should be incorporated into the Abstract, which states that “MyD88-/- PMNs failed to produce MitROS in response to pneumococcal infection” (line 25), which is not entirely reflective of the data presented in Fig 5.

Minor comments:

-Need to add statistical comparison of (WT + DPI) vs (WT + DPI + MitoTEMPO) in Fig 6A for statement in lines 227-228 stating that there was no additive effect of combining DPI with MitoTEMPO

-Pg 8, line 172-173. This Figure (3A) isn’t testing whether the PMNs are actively infected, more accurately whether they are binding to bacteria

-It would be useful to mention in the Discussion that the NLRP3 inflammasome can also induce MitROS, given that the potential bacterial triggers of this response are unresolved.

- Mention of LTA and lipopeptides should be included in the Discussion because they are specified in the Fig 13 model

-Show n.s. statistical result for Fig 3A graph

-Pg 6, line 116 change ‘damage’ to ‘damaged’

-Pg 2, line 23-24 ‘…pneumolysin but presence of live bacteria’ is missing ‘the’

-Pg 13, line 189 change ‘deactivate’ to ‘deactivated’

-Fig 6B shouldn’t have ‘+’ in between 19F and VC and 23F and VC to follow convention

established in other graphs

- ‘flowcytometry’ needs space between the two words

PLOS authors have the option to publish the peer review history of their article (what does this mean?). If published, this will include your full peer review and any attached files.

Reviewer #1: **Yes: **Jacqueline Kimmey

Reviewer #2: **Yes: **Sarah E Clark
---

## [Decision Letter · Decision Letter 1]

25 Oct 2022

Dear Dr. Bou Ghanem,

Thank you very much for submitting your manuscript "Mitochondrial ROS production by neutrophils is required for host antimicrobial function against Streptococcus pneumoniae and is controlled by A2B adenosine receptor signaling" for consideration at PLOS Pathogens. As with all papers reviewed by the journal, your manuscript was reviewed by members of the editorial board and by several independent reviewers. The reviewers appreciated the attention to an important topic. Based on the reviews, we are likely to accept this manuscript for publication, providing that you modify the manuscript according to the review recommendations.

The reviewers have very minor comments meant to enhance the clarity of the figures. Please address these. Otherwise, congratulations on excellent work and thank you for your responsiveness to the original reviewer concerns.

Sincerely,

Carlos Javier Orihuela, PhD

Associate Editor

PLOS Pathogens

Michael Wessels

Section Editor

PLOS Pathogens

Kasturi Haldar

Editor-in-Chief

PLOS Pathogens

orcid.org/0000-0001-5065-158X

Michael Malim

Editor-in-Chief

PLOS Pathogens

orcid.org/0000-0002-7699-2064

The reviewers have very minor comments meant to enhance the clarity of the figures. Please address these. Otherwise, congratulations on excellent work and thank you for your responsiveness to the original reviewer concerns.

Reviewer Comments (if any, and for reference):

Reviewer's Responses to Questions

**Part I - Summary**

Reviewer #1: As stated previously (in first review): This study investigates the role of mitochondrial ROS production from neutrophils in defense against Streptococcus pnuemoniae. This is a novel study in an important area of research, represents a significant amount of effort and the experimental data is of high quality. Concerns / weaknesses based on the first submission have been addressed in this revision with substantial new data as well as restructuring of the manuscript that made conclusions more clear and specific. Overall, this is a very nice study!

Reviewer #2: The revised manuscript addresses all of the concerns within the original manuscript, and includes new data (Fig 10) which significantly strengthen the connection between A2BR and MitROS in vivo. I have no further concerns regarding the publication of this study.

**Part II – Major Issues: Key Experiments Required for Acceptance**

Reviewer #1: none

Reviewer #2: (No Response)

**Part III – Minor Issues: Editorial and Data Presentation Modifications**

Reviewer #1: extremely minor comments:

Line 241: “Therefore, we investigated the role of the Gs-coupled A2B adenosine receptor in MitROS production by PMNs.” No rationale is provided to support the “therefore.” This sentence can be deleted or augmented with rational.

Re Fig 5A (line 248). It is still very hard to discern which line is which in this figure.

Line 321-323 / Fig 9C. Authors state there is more MitROS made by the A2BR-/- PMN, but in fig 9C, it doesn’t look striking – can you add the significance comparison here.

Line 336 / Fig 10. Most subfigures in this paper are arranged left to right on the top row, followed by the second row. This one is confusing as it is arranged vertically down a column followed by the second column.

Reviewer #2: (No Response)

PLOS authors have the option to publish the peer review history of their article (what does this mean?). If published, this will include your full peer review and any attached files.

Reviewer #1: **Yes: **Jacqueline Kimmey

Reviewer #2: **Yes: **Sarah E Clark

Figure Files:

Data Requirements:

Reproducibility:

References:

---

## [Editor Report · Decision Letter 2]

7 Nov 2022

Dear Dr. Bou Ghanem,

We are pleased to inform you that your manuscript 'Mitochondrial ROS production by neutrophils is required for host antimicrobial function against Streptococcus pneumoniae and is controlled by A2B adenosine receptor signaling' has been provisionally accepted for publication in PLOS Pathogens.

Best regards,

Carlos Javier Orihuela, PhD

Academic Editor

PLOS Pathogens

Michael Wessels

Section Editor

PLOS Pathogens

Kasturi Haldar

Editor-in-Chief

PLOS Pathogens

orcid.org/0000-0001-5065-158X

Michael Malim

Editor-in-Chief

PLOS Pathogens

orcid.org/0000-0002-7699-2064

Thank you for your willingness to address the reviewer's comments!
---

## [Editor Report · Acceptance letter]

11 Nov 2022

Dear Dr. Bou Ghanem,

We are delighted to inform you that your manuscript, "Mitochondrial ROS production by neutrophils is required for host antimicrobial function against Streptococcus pneumoniae and is controlled by A2B adenosine receptor signaling," has been formally accepted for publication in PLOS Pathogens.

Best regards,

Kasturi Haldar

Editor-in-Chief

PLOS Pathogens

orcid.org/0000-0001-5065-158X

Michael Malim

Editor-in-Chief

PLOS Pathogens

orcid.org/0000-0002-7699-2064